# Dynamic memristor-based reservoir computing for high-efficiency temporal signal processing

Yanan Zhong [1], Jianshi Tang [1,2 ✉], Xinyi Li[1], Bin Gao [1,2], He Qian[1,2] & Huaqiang Wu [1,2 ✉]

Reservoir computing is a highly efficient network for processing temporal signals due to its low training cost compared to standard recurrent neural networks, and generating rich reservoir states is critical in the hardware implementation. In this work, we report a parallel dynamic memristor-based reservoir computing system by applying a controllable mask process, in which the critical parameters, including state richness, feedback strength and input scaling, can be tuned by changing the mask length and the range of input signal. Our system achieves a low word error rate of 0.4% in the spoken-digit recognition and low normalized root mean square error of 0.046 in the time-series prediction of the Hénon map, which outperforms most existing hardware-based reservoir computing systems and also software-based one in the Hénon map prediction task. Our work could pave the road towards high-efficiency memristor-based reservoir computing systems to handle more complex temporal tasks in the future.

[1] Institute of Microelectronics, Beijing Innovation Center for Future Chips (ICFC), Tsinghua University, 100084 Beijing, China. [2] Beijing National Research Center for Information Science and Technology (BNRist), Tsinghua University, 100084 Beijing, China. ✉email: jtang@tsinghua.edu.cn; wuhq@tsinghua.edu.cn

In recent years, artificial neural networks (ANNs) have developed rapidly and played an important role in many different fields, such as object detection[1,2], natural language processing[3], autonomous driving[4], security[5], etc. Generally, ANNs can be loosely divided into two main categories depending on the network structure. One is feedforward neural networks in which the neurons are separated into layers and the signal only goes forward. There are many kinds of feedforward neural networks, including the well-known convolutional neural network[6], which are widely used to process static spatial patterns such as image recognition and object detection. However, this type of network may not be suitable for processing temporal signals because of the feedforward structure. The other kind of ANNs is recurrent neural network (RNN)[7,8] in which the neurons have recurrent connections. As a result, the history information of the input signal can be encoded into the internal states of the network so that short-term memory can be realized in this way. Therefore, RNN is capable of dealing with temporal tasks. Unlike feedforward neural networks, the training of RNN is usually very difficult and requires extensive computational power, mainly due to the problem of exploding or vanishing gradients in recurrent structures. In order to solve this problem, the concept of reservoir computing (RC) was proposed[9,10]. The main difference between RC and RNN is that in a RC network only the weights connected to the output layer need to be trained and the rest of the network remain fixed. As a result, the training process becomes linear and many simple training algorithms can be used, such as linear regression. At the software level, it has been shown that RC systems can achieve satisfactory performance in speech recognition[11], adaptive filtering[12], time series prediction[13,14], and many other fields[15]. For high system efficiency, many new materials and devices, such as spintronic oscillators[16,17], photonic modules[18–20], or memristors[21–24], have been studied for the hardware implementation of RC systems. Among them, remarkable progress has been made on memristors for the implementation of ANNs by taking advantage of their analog resistive switching properties[25–30]. Meanwhile, the inherent dynamic properties and nonlinear behavior of memristors also make them very suitable for the implementation of RC systems[31,32]. In a RC system, there are several key properties of the reservoir that largely affect the system performance, of which the richness of the reservoir states is one of the most important parameters. In the previous works, different reservoir states were usually generated using the inherent device-to-device variations[21,22]. Although this method can generate many reservoir states[33], the state richness is fixed after the devices are prepared and cannot be further adjusted in order to optimize the system performance. Besides, in these demonstrations, the memristor conductance was regarded as the reservoir state[21–23], so after each input signal, a read signal must be followed to read out the device conductance. This additional read operation would limit the speed of such RC systems.

In this report, we demonstrate a dynamic memristor-based RC system that uses a controllable mask process to generate rich reservoir states. By controlling the parameters in the mask process, we can adjust not only the state richness of the reservoir but also the feedback strength, both of which are critical properties that affect the RC system performance[34]. Besides, we directly use the memristor response to the input signal as the reservoir state, which can take advantage of the device nonlinearity and does not require additional read operations. Moreover, the nonlinear region of the system can be adjusted by simply changing the range of the input signal. By adjusting these system parameters, the implemented RC system can process temporal signal efficiently. Different temporal classification tasks of waveform classification and spoken-digit recognition are demonstrated in our RC system, where an extremely low normalized root mean square

error (NRMSE) of 0.14 and word error rate of 0.4% are achieved, respectively. Meanwhile, a time-series prediction task of the Hénon map is also performed in our system, and a low prediction error (NRMSE) of 0.046 is obtained, which is only half of the value obtained with a standard echo state network (ESN).

## Results

**Dynamic memristor-based RC system**. The dynamic memristor used in this work has a vertically stacked cross-point structure of $Ti/TiO_x/TaO_y/Pt$ (50 nm/16 nm/30 nm/50 nm), as schematically illustrated in Fig. 1a. The cross-sectional transmission electron microscope (TEM) image of the device is shown in Fig. 1b, and the corresponding elements distribution profile from energy-dispersive spectroscopy is shown in Fig. 1c. The details of device fabrication are described in the "Methods" section. The standard memristive $I–V$ hysteresis curves over multiple cycles are shown in Fig. 1d. The repeatable $I–V$ loops indicate a high stability and reliability of the device. Also, the $I–V$ curve is highly asymmetric under positive and negative voltage sweeps, which can be attributed to the Schottky barrier at the $TaO_y/Pt$ interface[35]. Such a strong nonlinearity of the dynamic memristor can be directly used to realize the activation function commonly used in ANNs. The dynamic characteristics of the device are also explored as shown in Fig. 1e. A write voltage pulse (amplitude of 3.0 V and pulse width of 1 ms) followed by several read voltage pulses (1.9 V, 10 μs) is applied on the device and the responding current is recorded for subsequent analysis. It can be seen from Fig. 1e that the current is integrated under the large write pulse (see Supplementary Fig. 1 for more detailed analysis) and then decays under the small read pulses, as the migration and diffusion of oxygen ions modulate the barrier height at the electrode/oxide interfaces[35]. The behavior of current decay over time is further analyzed in Fig. 1f, where a simple exponential relationship is used to fit the curve and the characteristic time $t_0$ obtained by fitting is about 400 μs. These experimental results imply that the output of the dynamic memristor is not only dependent on the current input but also relies on the history of the input signal[36,37]. Such short-term memory of the dynamic memristor gives it the ability to equivalently implement the neural network with recursive connections[34]. Combining the $I–V$ nonlinearity and short-term memory of the device, we realized a dynamic memristor-based RC system. As a comparison, Fig. 2a shows a conventional RC system that consists of three parts: input layer, reservoir, and output layer. The reservoir is the core of the RC system, which produces a large number of reservoir states that are very important for classification. Traditional approaches of making a reservoir use a network consisting of random connections of nonlinear neuron nodes. The interactions among neurons can remember the history information of the input signals and produce rich reservoir states. However, such RC architecture needs the random connections between multiple devices, which is very difficult for hardware implementation. In order to solve this problem, we incorporate the concept of time multiplexing and use a mask process to generate virtual nodes in time domain[34]. Through the dynamic and nonlinear response of the memristor, these virtual nodes are nonlinearly coupled to each other (see Supplementary Fig. 2). Figure 2b shows the schematic diagram of a dynamic memristor-based RC system based on this new architecture. First, the input signal is pre-processed through a time multiplexing procedure during which the input signal is multiplied by a mask matrix and then converted to a train of voltage pulses through a signal generation system. Every frame of the input signal can generate a pulse train with total length $\tau$ and pulse width $\delta$. Second, the pre-processed input is fed to the reservoir, which consists of a memristor connected in series with

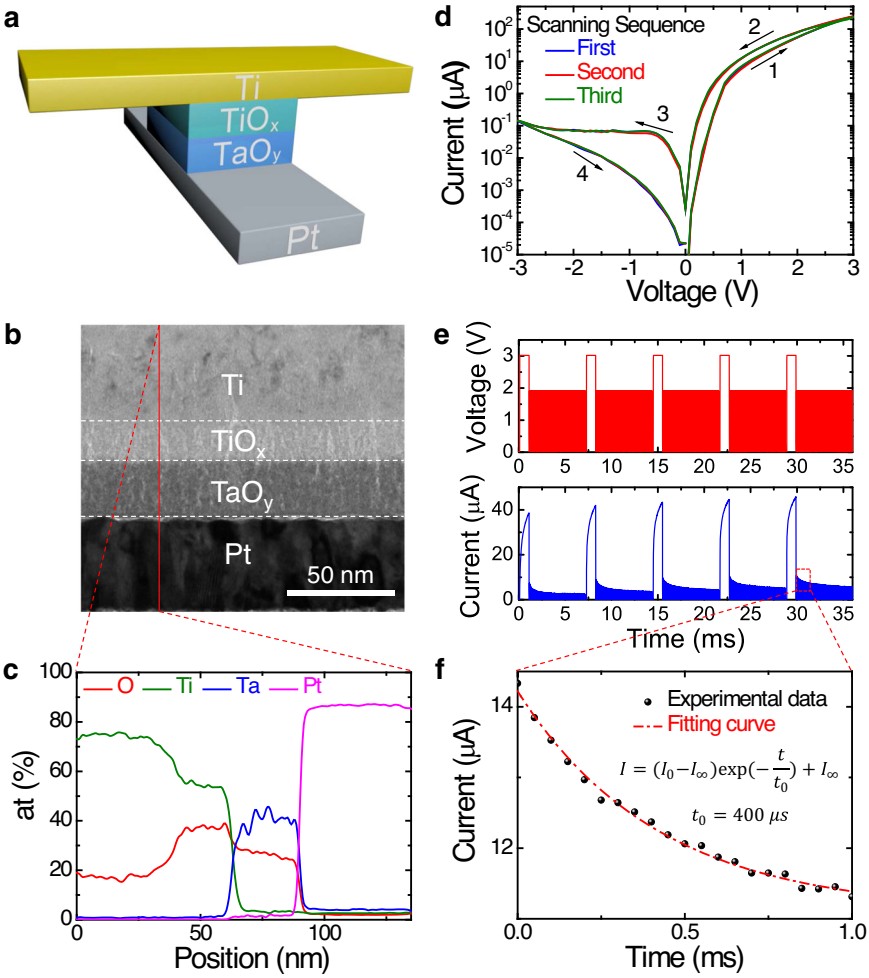

**Fig. 1 Device characteristics of dynamic memristor. a** Device structure and **b** cross-sectional transmission electron microscope (TEM) image of the fabricated dynamic memristor, consisting of a vertically stacked structure of Ti/TiO$_x$/TaO$_y$/Pt (50 nm/16 nm/30 nm/50 nm). **c** Corresponding elements distribution profile from energy-dispersive spectroscopy. **d** Device $I$–$V$ hysteresis curves. Three scans were repeated, and the arrows indicate the direction of the voltage scan. **e** The experiment exploring the dynamic characteristics of device. Here the input sequence is a periodic signal composed of a write voltage pulse (3.0 V, 1 ms) followed by several read voltage pulses (1.9 V, 10 μs) in one period. The responding current is recorded for subsequent analysis. **f** The current decay with time follows a simple exponential relationship and the characteristic time $t_0$ obtained by fitting is 400 μs.

a load resistor of $R_L = 4.7$ kΩ. The $R_L$ is used to convert the memristor output current to a voltage signal, which is then sampled as the reservoir states (that are the output of virtual nodes as shown in Fig. 1d). Finally, the output vector is a linear combination of the reservoir states and the weights are trained through linear regression. The details of the measurement set-up are described in the "Methods" section.

**Waveform classification**. In the above discussion, we proposed that a simple system connecting a dynamic memristor with a resistor can be regarded as a reservoir, which can generate a large amount of reservoir states for subsequent signal processing. In order to improve the system performance in practice, several single memristor-based reservoirs are connected in parallel to build a large parallel RC system as shown in Fig. 2c. A simple waveform classification task is used to test the temporal signal processing capability of our RC system[38,39]. As shown in Fig. 2d, the input sequence is a random combination of sine and square waveforms, and the desired output is the binary sequence that consists of 0 and 1 representing sine and square waveforms, respectively. To achieve the optimal classification results, we use ten reservoirs in parallel, where the mask (a one-dimensional

sequence with a length of four in this case) is different from each other. At the same time, the $I$–$V$ nonlinearity of dynamic memristor is directly used as the activation function as shown in Supplementary Fig. 3. In every time interval $\tau$, the output of RC system is the linear combination of all the reservoir states, where the weights are trained through simple linear regression method. NRMSE is used to measure the classification error[40], which is described as:

$$\text{NRMSE} = \sqrt{\frac{\left\langle \left\| y(t) - y_{\text{target}}(t) \right\|^2 \right\rangle}{\left\langle \left\| y_{\text{target}}(t) - \overline{y_{\text{target}}(t)} \right\|^2 \right\rangle}} \qquad (1)$$

where $y(t)$ is the output of RC system, $y_{\text{target}}(t)$ is the desired output, $\|\cdot\|$ denotes the Euclidean norm, and $\langle\cdot\rangle$ denotes the empirical mean. During the test, the lowest NRMSE we obtained is 0.14 and a typical result is also shown in Fig. 2d. In addition, we find that the length of the mask sequence has a critical influence on the performance of the RC system. As shown in Fig. 2e, the NRMSE of classification changes with the mask length $M$ when keeping the reservoir size the same $M \times N = 40$ ($N$ is the number of reservoirs in parallel). We can see that NRMSE becomes very

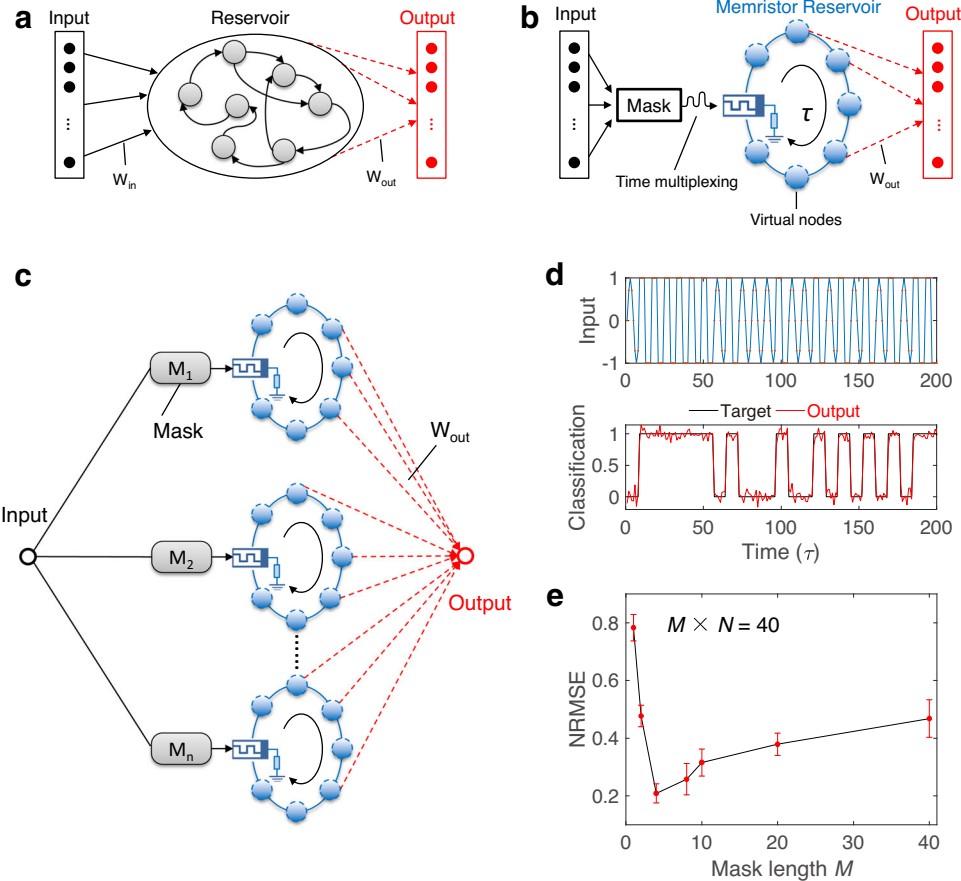

**Fig. 2 RC system architecture and waveform classification demonstration. a** Schematic of a conventional RC system. The input is fed to a reservoir, which is composed by a large number of nonlinear nodes. The internal connections among these nodes are random and fixed. The correct output learns from the states of nodes by training the output weights. **b** Schematic of the dynamic memristor-based RC system. For a given input, the input vector is transformed into a temporal signal through a mask (that is the time multiplexing process) and then fed to the reservoir, which consists of a dynamic memristor and a load resistor in series. The memristor responses within a duration time $\tau$ are selected as the virtual nodes with a fixed time step $\delta$. The output vector is a linear combination of the values in the virtual nodes and the weights ($\mathbf{W_{out}}$) can be trained through linear regression. **c** Schematic of a dynamic memristor-based parallel RC system, where the mask sequences are different for every single memristor RC unit. The output is the linear combination of all reservoir states. In our experiment, this parallel RC system is realized by testing single memristor in multiple cycles. **d** The input and classification result of sine and square waves. The input sequence is a random combination of sine and square waveforms, where the sampling points for each waveform are set to 8. The optimal classification results are achieved when the length of mask sequence and the number of reservoirs in parallel are set to 4 and 10, respectively, and the lowest NRMSE we get is 0.14. **e** NRMSE changes with the mask length when keeping the reservoir size (that is the product of mask length $M$ and number of reservoirs $N$) the same. Ten different devices are tested and the average of NRMSE reaches the minimum value as the mask length reaches 4. The error bar shows the variation between devices.

large when the mask length is either too long or too short and reaches the minimum value as the mask length is about 4. To explain such dependence on the mask length, let us consider two extreme cases with mask lengths of 40 and 1. When the mask length is as long as 40, the overall change of memristor conductance over duration $\tau$ would be large, which could easily drive the reservoir states to reach the upper or lower limit, thereby losing the ability to further process signals in the subsequent durations. In other words, the feedback strength between the two time durations decreases as the mask length increases, leading to a larger classification error. On the other hand, when the mask length is as short as 1, the binary combination of the mask sequence would be very limited, which limits the types of the mask sequence. In this case, the richness of the reservoir states in the parallel RC system is very low and the effective reservoir states could not support successful classification, leading to a large classification error as well. So in order to achieve the best classification result, the mask length needs to be carefully adjusted to make a trade-off between the feedback strength and the state

richness. In experiment, we find the optimal mask length to be around 4 that yields the lowest NRMSE of 0.14, which is lower than the previous value of 0.2 obtained with spintronic oscillator[16]. Further analysis of the effect of mask length on the feedback strength and state richness is discussed in Supplementary Figs. 4 and 5, where a method of using the peaks of the reservoir states in response to different input waveforms is developed to quantitatively analyze these two parameters. The test on cycle-to-cycle variation is shown in Supplementary Fig. 6. Another point worth mentioning is that the RC system is based on a single memristor (i.e., $N = 1$) when the mask length is 40. It can be seen from the experimental results that the parallel RC system has a better performance than the single memristor-based RC system by adjusting the mask length (e.g., $N = 10$ when $M = 4$), which not only increases the system speed but also reduces the error rate.

**Spoken-digit recognition**. To further evaluate the performance of dynamic memristor-based RC system on temporal classification

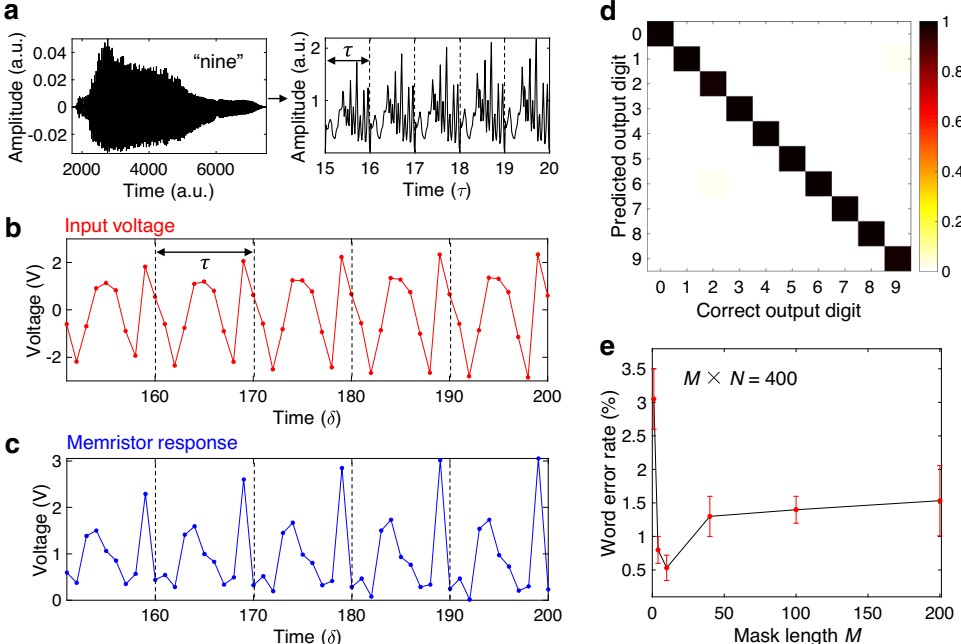

**Fig. 3 Spoken-digit recognition demonstration. a** Left: typical audio waveform of digit 9 pronounced by a female speaker. Right: cochlear spectrum (64 channels per frame) of the corresponding audio waveform. The channel values for each frame are transferred to the time domain with a duration of $\tau$. **b** Time multiplexing process. In each interval of duration $\tau$, the spectrum signal is multiplied by a mask matrix ($64 \times M$) containing randomly assigned binary values ($-1$ and $1$) to generate the input voltage sequence with a fixed time step $\delta$ ($\delta = 120 \, \mu s$) equal to $1/M$ of $\tau$, where $M$ is the mask length. The similar process repeats by $N$ times with different mask matrices in order to mimic $N$-parallel RC system. **c** During each time duration, the dynamic memristor response is recorded. The device current is first converted into voltage through the load resistor and then amplified and collected by the amplifier and ADC. After that, the $N$ times memristor responses in each duration of $\tau$ are combined into the reservoir states for subsequent classification. **d** Predicted results obtained from the memristor-based RC system versus the correct outputs, where the word error rate is as low as 0.4%. The two parameters $M$ and $N$ of the RC system are set to be 10 and 40, respectively. Color bar represents the normalized probability of each predicted result under the correct output. **e** Word error rate as a function of the mask length $M$, where the total reservoir size ($M \times N$) remains constant at 400. Similar to the waveform classification task, the average of word error rate reaches the lowest value when $M = 10$. The error bar represents the variation between devices.

tasks, the benchmark test of spoken-digit recognition is carried out using NIST TI-46 database. The input data are audio waveforms of isolated spoken digits (0–9 in English) pronounced by five different female speakers. The goal of spoken-digit recognition is to distinguish each digit independent of speakers. Therefore, feature extraction of audio signals is very important. Figure 3a–c illustrates the procedure of feature extraction of digit 9 based on the RC method. According to a standard procedure in speech recognition, the original audio waveform (resampled at 8 kHz) in Fig. 3a (left panel) is first filtered into a spectrum with 64 frequency channels per frame by using Lyon's passive ear model[41]. The channel values that represent the amplitude of the corresponding frequency for each frame are then transferred to the time domain with a duration of $\tau$ as shown in Fig. 3a (right panel). Figure 3b shows the pre-processed input signal after the mask process. Different from the previous waveform classification task, the mask here is a two-dimensional (2-D) matrix composed of randomly assigned binary values ($-1$ and $1$). In each interval of duration $\tau$, the spectrum signal is multiplied by a $64 \times M$ mask matrix to generate the input voltage sequence with a time step $\delta$ equal to $1/M$ of $\tau$, where $M$ is the mask length. The pre-processed input signal is then applied to the dynamic memristor, and the corresponding current is first converted to a voltage signal through the series resistor $R_L$ and then amplified and collected by the amplifier and analog-to-digital converter (ADC). The recorded memristor response is shown in Fig. 3c and the number of sampling points is set to be equal to $M$ per interval $\tau$. The time step is chosen as $\delta = 120 \, \mu s$, which must be shorter than the relaxation time $t_0$ (400 $\mu s$) of dynamic memristor. The mask and recording

processes are repeated $N$ times with different mask matrices in order to mimic $N$-parallel RC system. After that, the $N$ times memristor responses in each duration $\tau$ are combined into the reservoir states for subsequent classification.

The classification process contains two steps: training and testing. The 500 audio samples from TI-46 database are divided into two groups: 450 randomly selected samples for training and the rest 50 samples for testing. We use a ten-dimensional vector (target vector) to represent the classification result for the ten digits. For example, if the target digit is 9, the tenth number in the target vector will be 1 while the others should be 0. After feature extraction, the spoken digits are transformed into the reservoir states in each time interval $\tau$. The classification procedure is performed once at each interval and the final classification result is obtained from majority voting of the results at all intervals of one digit[11,16]. In an ideal situation, a correct classification can be given at each interval. We assume a weight matrix ($\mathbf{W_{out}}$) that can transform the reservoir states, which can be treated as an ($M \times N$)-dimensional vector, in each interval $\tau$ to the target vector. Therefore, the goal of the training process is to find a proper $\mathbf{W_{out}}$ for all the training samples to generate output vectors close to the corresponding target vectors. Here the linear regression method is used to calculate $\mathbf{W_{out}}$. We generate a target matrix $\mathbf{Y_{target}}$ by combining the target vectors at all the time intervals used for training. In the same way, we can also generate a response matrix $\mathbf{X}$ by combining the response vectors at all of the time intervals used for training. Subsequently, the weight matrix $\mathbf{W_{out}}$ is given by $\mathbf{W_{out}} = \mathbf{Y_{target}}\mathbf{X}^T(\mathbf{XX}^T)^{\dagger}$[42], where the symbol $\dagger$ represents Moore–Penrose pseudo-inverse.

During the testing process, the output vectors at all intervals of one digit are summed up. To obtain the final classification result, the element with the maximum value in the summed output vector predicts the corresponding digit (a winner-take-all method)[34]. To evaluate the accuracy, the recognition rate is defined as the percentage of correctly identified digits in all the testing digits. Furthermore, a tenfold cross-validation is used to ensure the reliability of the obtained recognition rate. To do that, the training and testing processes are repeated ten times and the data are randomly selected for training and testing for each time. The final recognition rate is the average of all the test results during tenfold cross-validation. Figure 3d shows the predicted digits obtained from the memristor-based RC system versus the correct digits, where the color depth is proportional to the number of correctly classified digits. The word error rate is as low as 0.4% (i.e., recognition rate of 99.6%) when $M$ and $N$ are set to be 10 and 40, respectively, which is lower than the value of 0.8% obtained by the memristor-based RC system in the previous work[22]. In Fig. 3e, the dependence of the word error rate on the mask length is investigated, where the total reservoir size ($M \times N$) remains constant at 400. Similar to the previous waveform classification task, the word error rate increases when the mask length is too long or too short. It can be seen from the experimental data that the lowest average word error rate is achieved when the mask length is about ten. In addition, the effect of the reservoir size on the RC system has also been studied, and the experimental result is shown in Supplementary Fig. 7. It is found that the word error rate decreases with the reservoir size, because a larger reservoir can create more reservoir states and hence retain more features of the input signals.

**Time-series prediction**. In addition to the classification of temporal signals in the above two demonstrations, we also perform another benchmark task to demonstrate the prediction of temporal signals. Hénon map has been established as a typical discrete-time dynamic system with chaotic behavior[43]. It describes a nonlinear 2-D mapping that transforms a point ($x(n)$, $y(n)$) on the plane into a new point ($x(n + 1)$, $y(n + 1)$), defined as follows:

$$x(n + 1) = y(n) - 1.4x(n)^2 \qquad (2)$$

$$y(n + 1) = 0.3x(n) + w(n) \qquad (3)$$

where $w(n)$ is a Gaussian noise with a mean value of 0 and a standard deviation of 0.05. The task is to predict the system position at time step $n + 1$, given the values up to time step $n$. The system can be described as an equation containing only $x$ if we combine Eqs. (3) and (2), so the input of the task is $x(n)$ and the target output is $x(n + 1)$. Using these equations, we generate the Hénon map dataset with a sequence length of 2000, in which the first 1000 data points is used for training and the rest is used for testing. To execute this task in our memristor-based parallel RC system, the input time series $x(n)$ is linearly mapped to the voltage range of [$V_{min}$, $V_{max}$]. The mask process is similar to the one used in the previous waveform classification task. During each time interval $\tau$, the pre-processed signal is multiplied by a special mask with a length of $M$ to generate the input voltage sequence with a time step $\delta$ ($\delta = 120\ \mu s$). An $N$-parallel RC system is realized by using different mask sequences. The training and testing processes are similar to the previous tasks and the only difference is that a bias is added to the output layer to neutralize the influence of input signal offset on the output. Both bias and weights are trained with linear regression. After finding the suitable parameters, our RC system can achieve excellent performance on the time-series prediction. For example, Fig. 4a shows the predicted time series versus the ideal target during the testing process for the first 200 time steps, where a very low NRMSE of 0.046 is achieved by the dynamic memristor-based RC system. Here the parameters are set to be $M = 4$, $N = 25$, $V_{max} = 2.5\ V$, and $V_{min} = -0.8\ V$. In order to show the predicted results more intuitively, Fig. 4b is a 2-D display of the Hénon map in Fig. 4a, which demonstrates that the strange attractor of the Hénon map can be well reconstructed.

As mentioned above, the parameter setting has a big impact on the performance of the memristor-based RC system. As shown in Fig. 4c, the output of our RC system has a relatively large prediction error with an NRMSE of 0.14, when changing $M$ and $V_{max}$ to 25 and 2.0 V, respectively, while keeping $V_{min}$ and the total reservoir size ($M \times N$) the same. Furthermore, a systematic experiment is conducted and the results are shown in Fig. 4d, where the system performance varies with the two parameters of $M$ and $V_{max}$. Here the parameter $V_{max}$ is related to the input scaling, which has been proven to be an important parameter that affects the performance of RC system[34]. Different input scalings are realized by simply changing $V_{max}$ while setting $V_{min}$ to be a fixed value close to 0. In the experiment, $V_{min}$ is empirically set to be a small negative value ($-0.8$ V) in order to balance the resistive state of the dynamic memristor. It can be seen from Fig. 4d that the prediction error (NRMSE) varies with not only $M$ but also $V_{max}$ obviously. The best performance is achieved when $M = 4$ and $V_{max} = 2.5$ V, as too large or too small $M$ and $V_{max}$ would cause relatively poor prediction results. Similar experimental results are obtained by testing on different devices as shown in Supplementary Fig. 8. The effect of mask length has been analyzed in the previous sections. Here we further study the influence of $V_{max}$ on the performance of the memristor-based RC system. The value of $V_{max}$ determines the nonlinear region of the device in response to the input signal. As shown in Supplementary Fig. 9a, the response of the dynamic memristor to the input voltage has an apparent threshold. The region around the threshold has a strong nonlinearity, while the region far away from the threshold has a weaker nonlinearity. If $V_{max}$ is too small, the resistance state of the device is difficult to be changed (see Supplementary Fig. 9b), which would lead to poor system performance. However, if $V_{max}$ is too large, the overall nonlinearity in the entire input region would be reduced, which also degrades the RC system performance. Therefore, in order to achieve the best system performance, the value of $V_{max}$ needs to be carefully adjusted.

In addition, a comparison of the prediction error versus reservoir size between the software- and memristor-based RC systems is shown in Fig. 4e. The lowest prediction error achieved by our dynamic memristor-based RC system (NRMSE = 0.046) is only half of the value achieved by a standard ESN system (NRMSE = 0.091) as reported in previous work[40], and the total reservoir size used in our RC system is also half of that in the standard ESN system. It is worth mentioning that the prediction error of ESN used for comparison here is the state-of-the-art value that a single-layer RC system can achieve, and lower error can be obtained when using multi-layer RC systems with more complex training process[44]. For comparison, the simulation result using a simple dynamic memristor model is also shown in Fig. 4e, where the prediction error achieved by simulation is much lower than that achieved by experiment and is close to the values achieved by multi-layer RC systems. The simulation details are described in Supplementary Fig. 10 and Supplementary Table 1. These results suggest that the dynamic memristor-based parallel RC system that we proposed in this work still has room for performance optimization.

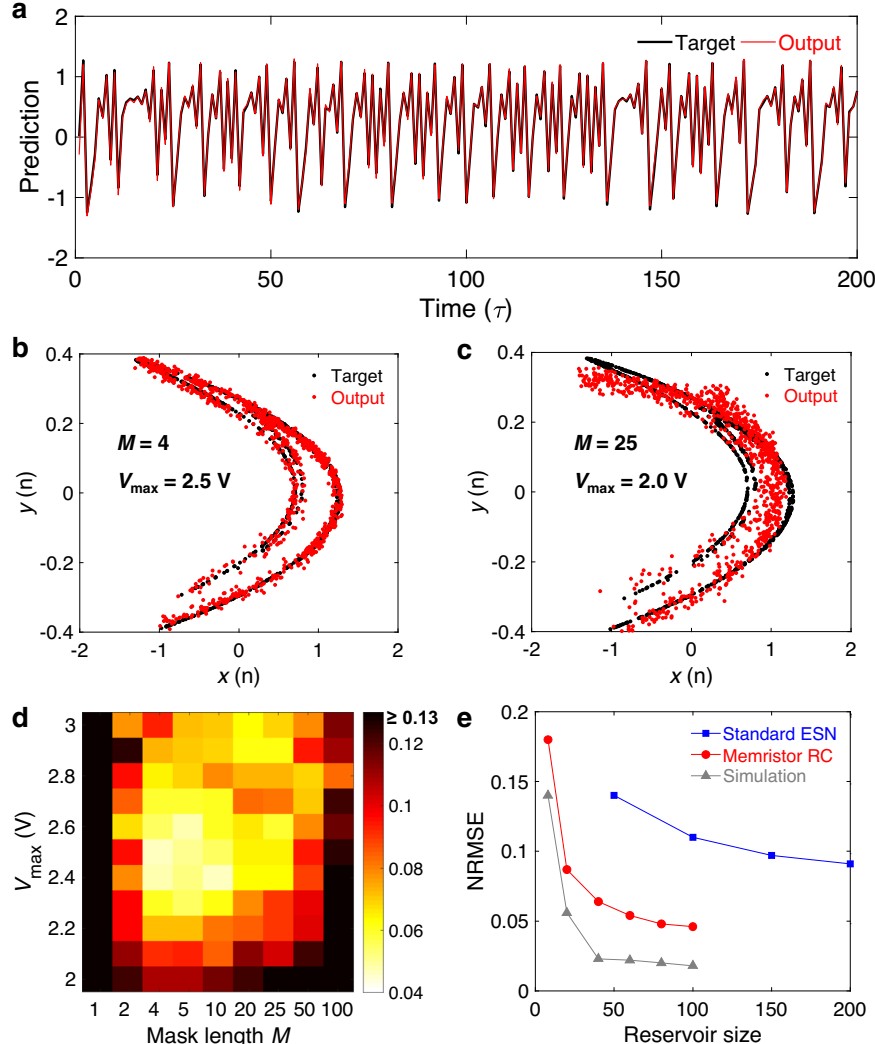

**Fig. 4 Demonstration of Hénon map prediction. a** The predicted results obtained by the memristor-based parallel RC system, where the black line represents the ideal target and the red line represents the experimental output from the RC system. Test parameters are set to be $M = 4$, $N = 25$, $V_{max} = 2.5$ V and $V_{min} = -0.8$ V. **b**, **c** 2-D display of the predicted results under different test parameters, where the $M$ and $V_{max}$ in **b** and **c** are 4, 2.5 V and 25, 2.0 V, respectively. $V_{min}$ and the total reservoir size ($M \times N$) remain unchanged at $-0.8$ V and 100, respectively. **d** The prediction error varies with the two test parameters $M$ (1–100) and $V_{max}$ (2.0–3.0 V), where the color bar represents the NRMSE (values >0.13 are shown in black in the figure). $V_{min}$ and $M \times N$ remain unchanged. **e** The NRMSE changes with the reservoir size in different RC systems including the standard ESN (results in ref. [40]), memristor-based parallel RC, and software-simulated one. In both experiment and simulation, the mask length remains constant at 4, and the total reservoir size is adjusted by changing the number of parallel reservoirs $N$.

## Discussion

In summary, a high-performance parallel RC system has been realized using a novel Ti/TiO$_x$/TaO$_y$/Pt dynamic memristor. By applying a simple mask process, we show that even a single dynamic memristor can be treated as a reservoir, which is subsequently used to build a parallel RC system. By choosing the appropriate mask length and the range of input voltage, our RC system can process temporal signals efficiently. Low NRMSE and word error rate of 0.14 and 0.4% have been achieved for the waveform classification and spoken-digit recognition, respectively, and meanwhile the prediction error of the Hénon map task is as low as 0.046, which is almost 50% less than the value obtained by a standard ESN system. Furthermore, the spatial signal processing task of handwritten-digit recognition is also demonstrated by our RC system as shown in Supplementary Fig. 11, where a high recognition accuracy of 97.6% is achieved and the accuracy loss is just 0.4% compared to the software baseline. Compared with the previous work[22], the operating power of our memristor-based RC system is much lower owing to the mask process (see Supplementary Table 2), and the energy consumption can be further reduced by reducing the input voltage pulse width. The parallel RC system in this work is implemented on a single memristor running in serial mode, which is very compact and efficient, proving the feasibility and high efficiency of memristor-based RC system. To further enable parallel processing of input signals and increase the complexity of the RC system, a more sophisticated RC system based on multiple memristors with inner connections (see Supplementary Fig. 12 for the diagram of a conceived multi-layer memristor-based RC system) will be constructed in the future.

## Methods

**Device fabrication**. The dynamic memristor device was fabricated as a cross-point structure on a silicon substrate with 200 nm thermally grown silicon oxide on it. First, inert metal Pt was deposited and patterned on the substrate as the bottom electrode. The thickness and width of the bottom electrode are 50 nm and 10 µm, respectively. Then the functional 30 nm-thick TaO$_y$ and 16 nm-thick TiO$_x$ oxide

layers were deposited by the reactive sputtering method with Ar and $O_2$ mixed atmosphere[45]. Finally, the top electrode Ti was deposited and patterned with the same thickness and width as the bottom electrode.

**Measurement set-up**. The basic electrical behaviors of the dynamic memristor were characterized at room temperature in a probe station connecting to a semiconductor parameter analyzer (Agilent B1500). The thickness of each layer of the device was verified by TEM. The experimental RC system is realized with the cooperation of personal computer (PC), microcontroller unit (MCU) with peripheral circuits, and memristor device. The PC is used to run the basic loop of RC algorithm, which is realized by MATLAB code. The MCU used in our experiment is STM32 with 12-bit digital-to-analog converter (DAC) and ADC modules. The peripheral circuits consist of input and output amplifiers. The function of STM32 and amplifier is to connect the PC with the memristor device. Take the spoken-digit recognition task for example. The PC pre-processes the spoken signal into a discrete sequence of real numbers between −1 and 1. This data sequence is transferred to the buffer of STM32 through UART communication. The DAC module of STM32 then generates voltage pulses with pulse width of 120 μs and amplitude (0–3.3 V) corresponding to data values. The input amplifier resizes the amplitude of voltage pulse between −3 to 3 V and applies it to the memristor device. The constant $R_L$ in series with the memristor is used to convert the response current into a voltage signal. The value of $R_L$ is dependent on the magnitude of the current response $I_{memristor}$ and the maximum gain of the amplifier ($A_v = 1000$) we used. In the speech recognition task, our system need to detect a current on the order of 1 μA. As the voltage upper limit of our ADC is $V_{ADC} = 3.3$ V, the load resistor should satisfy the following equation: $\frac{V_{ADC}}{I_{memristor} \times R_L} \leq A_v$. In our system, we have $R_L \geq \frac{V_{ADC}}{I_{memristor} \times A_v} \approx 3.3$ kΩ. In addition, in order to reduce the voltage drop on the load resistor connected in series with the dynamic memristor, $R_L$ should be much smaller than the memristor resistance (7 MΩ–20 kΩ measured in voltage range of 1–3 V). As a result, the value of $R_L$ is chosen to be 4.7 kΩ in our experiment. The output amplifier transforms the small current signal of memristor into a large voltage signal (0–3.3 V), which is then sampled by the ADC module. Finally, the ADC data are transferred from STM32 back to the PC for post-processing. The simulations of dynamic memristor-based RC and software-based RC are both implemented in MATLAB.

## Data availability

The data that support the findings of this study are available from the corresponding author upon reasonable request. Source data are provided with this paper.

## Code availability

The code that supports the dynamic memristor-based RC simulations in this study is available at https://github.com/Tsinghua-LEMON-Lab/Reservoir-computing/ (https://doi.org/10.5281/zenodo.4299344). Other codes that support the findings of this study are available from the corresponding authors upon reasonable request.

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

## Acknowledgements

This work was supported in part by China key research and development program (2019YFB2205403) and Natural Science Foundation of China (61974081, 91964104, 61851404, 61674089).

## Author contributions

Y.Z. and J.T. conceived and designed the experiments. X.L., B.G., and H.Q. contributed to the device preparation and material analysis. Y.Z. performed the experiments and data analysis. Y.Z. and J.T. wrote the paper. All authors discussed the results and commented on the manuscript. J.T., H.W., and H.Q. supervised the project.

## Competing interests

The authors declare no competing interests.
