## [Peer Review File · Nature Communications]

REVIEWER COMMENTS

Reviewer #1 (Remarks to the Author):

This manuscript builds on previous work using the dynamical properties of a memristor to serve as the basis of a reservoir in a reservoir computing (RC) neural network. Specifically, they suggest improvements to the memristor-RC system proposed by Moon et al (Nat. Electron. 2, 480, 2019, ref 22 in the manuscript) where a single $\text{TiO}_x/\text{TaO}_y$ bilayer memristor is time-multiplexed to act as the entire reservoir. The utility of the modified design is demonstrated on two main tasks: spoken digit recognition and handwritten digit recognition.

The topic of RC processing using memristors is of interest to the community, and the key differentiating claim of time multiplexing with a single memristor does appear to be a viable method of implementing this algorithm. This manuscript is reasonably clear in stating the claimed novelty - which is appreciated. However, I find this work to be a relatively incremental change from the work in Moon et al, and the claimed improvements over previous are not fully supported for reasons I will discuss below. For this reason, I cannot recommend this particular manuscript for publication in Nature Communications.

The following provides some detailed issues to consider if a future version of the manuscript is prepared.

The first key claimed improvement over previous work (Moon et al/ref 22) is that "it is difficult to control the variation between devices and hence it is now well reproducible." After examining ref 22, I agree this is correct that the previous work uses the variation between devices to produce the RC states for different nodes (although in each node, the dynamics of a single device is used). The challenge for the previous work, which is discussed in that paper, is to duplicate the reservoir characteristics in a separate piece of hardware with different devices - which is necessary for the implementation to have practical impact. The single-device system proposed in the current manuscript, however, does not solve this challenge. If the single device system is transferred to a separate piece of hardware, again with a new device, the new system would be subject to the same device-to-device variability. So I do not see how this manuscript solves the previous weakness. More specifically, I assume in the present scheme, if a new device is used, all of the output weights (in Fig 2) will have to be retrained to accommodate the new device/reservoir characteristics.

The second claimed improvement is related to a weakness in the previously published technique described as: "after each input signal, a read signal must be followed to read out the device conductance. This additional read operation would limit the speed of such RC systems." While I agree that this read step is required in the previous design, it seems that the new proposed design in the present manuscript would have significant delays due to the sequential testing of a single memristor for each time-multiplexed node. Hence it is not clear, nor is it quantified that the proposed new system would have a reduced latency over the previous system. Also the characteristic time for the bilayer device and time steps used in this system are relatively long, so I assume this would contribute to a higher latency of the recognition system based on the new design. Furthermore, both the claims of "high efficiency" and "high performance" are made in this manuscript, but are only justified qualitatively, and enough information is not provided for the reader to quantify this advantage. If a future version of this manuscript is prepared, it would make it significantly more compelling if the authors could rigorously quantify the advantages (in terms of latency, energy, or other relevant attributes) over previous work, such as the advantage over the system proposed and quantified in ref. 22.

The spoken digit classification is a useful example to demonstrate the accuracy of the memristor RC system. However, Fig 3(d) does not convey meaningful information about the accuracy because the color scale ends at 50% correct and clearly all of the data variation is in the range of above 50%. I would recommend rescaling or removing that plot.

The example of MNIST being performed on the RC net is of much less interest, as handwritten images are not a dynamic system that requires an RC net, and this is a relatively easy and efficient system to solve on a standard multi-layer perceptron or convolutional neural network. I would recommend replacing this with a more relevant example.

Reviewer #2 (Remarks to the Author):

This paper discusses a dynamic memristor based reservoir computing (RC) in processing spatial and temporal signals by choosing a certain mask length and the number of reservoirs. The authors show the simulation results through low NRMSE and word error rate for waveform classification and spoken-digit recognition and has a high recognition rate for handwritten-digit recognition. The use of memristor in realizing this memristor-based RC is well explained.

This work also has a potential impact to the field since the RC system does not use additional read operations and uses the device nonlinearity from the memristors. This makes the system very efficient. The described system is able to handle spatial-temporal tasks very efficiently. Overall, this paper gives a solid explanation of the system with good results compared to software-based RC system and shows compact, efficient and feasible results with memristor.

This paper could be further strengthened if the authors could address the following questions and concerns:

- 1) This Memristor-Based RC system paper states that to enable parallel processing of input signals a more sophisticated RC system can be used. Here the experiment was realized using a single memristor in multiple cycles so more information could be included for the process of making a more sophisticated RC.
- 2) The process for determining the mask length can also be discussed in detail since it is a key feature for this paper.
- 3) An explanation could be added as to how the value of the load resistor was calculated.
- 4) More information about the memristor could be added about the device characteristics of the memristor such as the critical current and the reason for the choice of elements used.

Reviewer #3 (Remarks to the Author):

The authors present a dynamic memristor-based reservoir computing approach and use it for solving spatiotemporal signals. The article fails to cite key background literature, does not provide sufficient details, and most importantly, does not compare the results with related/state-of-the-art work. The

only comparison the authors provide is with a software-based implementation. However, there's a significant body of work on memristor-based reservoirs today. The authors should have compared the performance of their proposed approach with the existing literature, not just with a software-based approach. Without a comparison with other memristor-based RC systems, a reader will be unable to evaluate the value and the impact of what the authors propose.

Other comments (in no particular order):

The authors use many vague terms, such as "not well," "almost equivalent," "for a while," etc. These must be replaced by quantitative statements.

The concept of the "virtual nodes" is not motivated and described in detail.

The authors fail to cite the first memristive RC paper from 2012:
<https://ieeexplore.ieee.org/document/6464167>

"64: [...] to control the variation between devices and hence it is not well reproducible." Well, isn't the goal of variation precisely to provide rich dynamics that do not need to be reproducible? That's the entire point of RC. The training of the output layer will deal with that "problem," which is not really a problem. In fact, it's generally good to have variation. See for example:
<https://ieeexplore.ieee.org/document/6623028/>

"67. where an extremely low normalized root mean square error (NRMSE) of 0.14 and word error rate of 0.4% are achieved respectively." Compared to what?

Well, I do not think that Nature readers need to be reminded of the NRMSE formula.

Abstract:

- "[...] however, this method is not well controllable and reproducible."  It is not clear why the authors are saying that. This should be clarified even in the abstract.

- "The performance of dynamic memristor-based RC system is almost equivalent to the software-based one" -> The authors need to quantify that statement. How much better? E.g., how many %?

"37. As a result, input signal can stay in the network for a while so that short-term memory can be realized in this way"  This sentence is unclear. I intuitively know what the authors mean, but the statement needs to be rewritten and clarified.

Response Letter to Reviewers' Comments

We sincerely appreciate the valuable time the reviewers have spent reviewing our manuscript and providing insightful comments and suggestions to help further improve the quality of our work. We have carried out additional experiments and revised the manuscript thoroughly to address all of the reviewers' comments. Our point-by-point responses to the reviewers' comments are as follows.

Reviewer #1 (Remarks to the Author):

This manuscript builds on previous work using the dynamical properties of a memristor to serve as the basis of a reservoir in a reservoir computing (RC) neural network. Specifically, they suggests improvements to the memristor-RC system proposed by Moon et al (Nat. Electron. 2, 480, 2019, ref 22 in the manuscript) where a single TiOx/TaOy bilayer memristor is time-multiplexed to act as the entire reservoir. The utility of the modified design is demonstrated on two main tasks: spoken digit recognition and handwritten digit recognition. The topic of RC processing using memristors is of interest to the community, and the key differentiating claim of time multiplexing with a single memristor does appear to be viable method of implementing this algorithm. This manuscript is reasonably clear in stating the claimed novelty - which is appreciated. However, I find this work be a relatively incremental change from the work in Moon et al, and the claimed improvements over previous are not fully supported for reasons I will discuss below. For this reason, I cannot recommend this particular manuscript for publication in Nature Communications.

Response:

We thank the reviewer for recognizing the importance of using a newly developed dynamic memristor to implement reservoir computing (RC) and the differentiating method of time multiplexing developed in this work. Our detailed responses to your technical comments are provided below.

Some concerns are listed below:

Comment #1

(1) The first key claimed improvement over previous work (Moon et al/ref 22) is that "it is difficult to control the variation between devices and hence it is now well reproducible." After examining ref 22, I agree this is correct that the previous work uses the variation between devices to produce the RC states for different nodes (although in each node, the dynamics of a single device is used). The challenge for the previous work, which is discussed in that paper, is to duplicate the reservoir characteristics in a separate piece of hardware with different devices - which is necessary for the implementation to have practical impact. The single-device system proposed in the current manuscript, however, does not solve this challenge. If the single device system is transferred to a

separate piece of hardware, again with a new device, the new system would be subject to the same device to device variability. So I do not see how this manuscript solves the previous weakness. More specifically, I assume in the present scheme, if a new device is used, all of the output weights (in Fig 2) will have to be retrained to accommodate the new device/reservoir characteristics.

Response:

Thank you for your comment. We apologize that our original statement may be a little misleading, and we have rephrased it to be more accurate in expression. Ref. 22 is a pioneer work that demonstrated the feasibility of using dynamic-memristor-based RC system to process temporal tasks with high accuracy. As the reviewer pointed out, there is a common issue for memristor-based RC systems, including both ref. 22 and our work, that the output weights would need to be retrained to achieve the best system performance if a new device is used. This is similar to the training process in typical memristor-based artificial neural networks when new synaptic devices are used. However, one key challenge of RC that we want to address in our work is that **many critical parameters of the RC system, such as the state richness and feedback strength of the reservoir, cannot be further adjusted in the previous work of ref. 22.** These parameters of the reservoir have significant impacts on the RC system performance (e.g. Appeltant L. et al., *Nat. Commun.*, 2011; Larger L. et al., *Optics Express*, 2012), so if they cannot be adjusted, then there could be a situation in which no matter how we train the output weights, the system still has a poor performance. Besides, the desired values of these parameters generally depend on the specific tasks. Therefore, **it is necessary to represent these parameters using the system properties that can be externally adjusted,** similar to the case in software-based RC systems where parameters like the input weight scaling, reservoir sparsity and spectral radius of the weight matrix are usually selected and optimized to represent the overall properties of the reservoir (e.g. Rodan A. et al., *IEEE Trans. Neural Netw.*, 2011; Gallicchio C. et al., *Neurocomputing*, 2018; Jaeger H. et al., *GMD-Forschungszentrum Informationstechnik*, 2002., 2002). Inspired by this idea, in this work **we proposed to use the mask length and input signal range as adjustable parameters to further tune the properties of the reservoir and achieve the best RC system performance, as experimentally demonstrated in Figure R1 below.** The device-to-device variation represented by the error bar in **Figure R1** also proves that our RC system is reproducible when using different dynamic memristors for the system implementation. By controlling the parameters in the mask process, we can adjust not only the state richness of the reservoir but also the feedback strength both of which are critical properties that affect the RC system performance. **This is one of the key novelties of our work beyond ref. 22.**

To clarify this point, we have added **Figure R1** as **Supplementary Figure S4** in the revised **Supplementary Information** and also incorporated the above discussions in the revised manuscript:

On pages 1, lines 20 – 21: “**however, the state richness created by such method is**

fixed after the devices are prepared, and cannot be further adjusted in order to optimize the system performance.”

On pages 3, lines 64 – 70: “In a RC system, there are several properties of the reservoir that largely affect the system performance, of which the richness of the reservoir states is one of the most important parameters. In the previous works, different reservoir states were usually generated using the inherent device-to-device variations. Although this method can generate many reservoir states, the state richness is fixed after the devices are prepared, and cannot be further adjusted in order to optimize the system performance.”

On pages 3 – 4, lines 75 – 78: “By controlling the parameters in the mask process, we can adjust not only the state richness of the reservoir but also the feedback strength both of which are critical properties that affect the RC system performance.”

Figure R1. (a) The feedback strength and state richness changes with the mask length M , where the feedback strength and state richness are quantified as $\overline{\Delta V_p}$ and $\sigma(V_{p1})$ respectively. $\overline{\Delta V_p}$ is the average of the five largest ΔV_p in all reservoir states and $\sigma(V_{p1})$ is the standard deviation of V_{p1} that is the peak response voltage of the sine wave. We can see that the feedback strength decrease with the mask length while the state richness increases with the mask length and saturates when the mask length is larger than 4. The best performance of the RC system is achieved when the mask length is around 4 that yields a trade-off between the feedback strength and the state richness. (b) The classification error (NRMSE) changes with the mask length, and NRMSE reaches the minimum value of 0.14 when the mask length is 4. The error bars in both (a) and (b) represent the variations between devices.

Comment #2

(2) The second claimed improvement is related to a weakness in the previously published technique described as: "after each input signal, a read signal must be followed to read out the device conductance. This additional read operation would limit the speed of such RC systems." While I agree that this read step is required in the previous design, it seems that the new proposed design in the present manuscript would have significant delays due to the sequential testing of a single memristor for each time-multiplexed node. Hence it is not clear, nor is it quantified that the proposed new system would have a reduced latency over the previous system. Also the characteristic time for the bilayer device and time steps used in this system are relatively long, so I assume this would contribute to a higher latency of the recognition system based on the new design. Furthermore, both the claims of "high efficiency" and "high performance" are made in this manuscript, but are only justified qualitatively, and enough information is not provided for the reader to quantify this advantage. If a future version of this manuscript is prepared, it would make it significantly more compelling if the authors could rigorously quantify the advantages (in terms of latency, energy, or other relevant attributes) over previous work, such as the advantage over the system proposed and quantified in ref. 22.

Response:

Thank you for your comment. In this work, the length of a unit time step for spoken-digit recognition task is about 1.2 ms ($10 \times 120 \mu\text{s}$) which is longer than the value of $250 \mu\text{s}$ reported in the previous work of ref. 22. The higher latency of our system is because that the pulse width ($120 \mu\text{s}$) of our RC system is much larger than $10 \mu\text{s}$ in the previous system. We shall point out that **the use of such a large pulse width is due to the hardware limitation (such as the bandwidth of the amplifiers) of our test system which can be readily improved in the future.** To further clarify this, we have evaluated the minimum pulse width we could use in our RC system. As shown in **Figure R2a**, the current response under different voltage pulse amplitudes has been studied. It can be seen that the increase of current shows two different regimes: when the voltage is just applied, the current shows a very fast increase and then the increase becomes much slower after a few microseconds. The characteristic time in the fast-increase regime determines the fastest pulse width we can apply on the device. In order to obtain the characteristic time, a double exponential function is used to fit the experimental data. As shown in **Figure R2b**, **the characteristic time t_a decreases when the voltage amplitude increases, and t_a becomes less than $10 \mu\text{s}$ when the applied voltage is greater than 2.4 V. That means the pulse width ($120 \mu\text{s}$) of the input signal in our RC system can be further reduced to $10 \mu\text{s}$, which could shorten the length of a unit time step in our RC system to be below $100 \mu\text{s}$, much smaller than the value of $250 \mu\text{s}$ in the previous work.**

As to the mask process used in our RC system, although the time multiplexing step could indeed increase the latency, **it brings more benefits by adjusting the reservoir**

properties (e.g., increasing the reservoir state richness) to further improve the system performance. Furthermore, the increased latency caused by the mask process is controllable. We can reduce the mask length and increase the number of parallel devices to minimize the system latency. Hence, **for different tasks, we can trade off the latency, performance and area of our RC system by adjusting the mask length.**

As the reviewer suggested, we have further estimated the power consumption of our RC system. In this work, because of the mask process, both the input voltage and the corresponding current vary within a certain range. Thus, the power consumption of processing a single input can be estimated by calculating the average power for all the inputs in a task. For the spoken-digit recognition task, the input voltage mostly varies from -2 to 2 V (the corresponding current is in the range of -0.05 to 50 μA), so **the average power is then roughly estimated to be** $(-2 \text{ V} \times -0.05 \mu\text{A} + 2 \text{ V} \times 50 \mu\text{A}) \div 2 \approx 50 \mu\text{W}$ **for our RC system.** In the previous work, the power consumption for processing a single input is approximately $3.0 \text{ V} \times 100 \mu\text{A} = 300 \mu\text{W}$, which is 6 times higher than the value estimated in this work. A brief comparison is made in **Table R1.**

To clarify this point, we have added **Table R1 and Figure R2** as **Supplementary Table S2 and Supplementary Figure S2** respectively in the revised **Supplementary Information** and also incorporated the above discussions in the revised manuscript:

On pages 7, lines 173 – 175: “**In experiment, we find the optimal mask length to be around 4 that yields the lowest NRMSE of 0.14, which is lower than the previous value of 0.2 obtained with spintronic oscillator.**”

On pages 10, lines 236 – 239: “**The word error rate is as low as 0.4% (i.e., recognition rate of 99.6%) when M and N are set to be 10 and 40 respectively, which is lower than the value of 0.8% obtained by the memristor-based RC system in previous work.**”

On pages 12, lines 303 – 307: “**The lowest prediction error achieved by our dynamic memristor-based RC system (NRMSE = 0.046) is only half of the value achieved by a standard ESN system (NRMSE = 0.091) as reported in previous work, and the total reservoir size used in our RC system is also half of that in the standard ESN system.**”

RC System	Power per input	Performance
Previous work (ref. 22)	300 μW	99.2%
This work	50 μW	99.6%

Table R1. The estimated power and performance of different memristor-based RC systems in the spoken-digit recognition task. In previous work (e.g. Moon J. et al., *Nat. Electron.*, 2019), the power consumption for processing a single input is approximately $3.0 \text{ V} \times 100 \mu\text{A} = 300 \mu\text{W}$, where the values of voltage and current are both taken from the reference. In this work, because of the mask process, both the input voltage and the corresponding current vary within a certain range. Thus, the

power consumption of processing a single input should be obtained by calculating the average power for all the inputs in a task. For the spoken-digit recognition task, the values of input voltage mostly vary from -2 to 2V (The corresponding current is in the range of -0.05 to 50 μA), so the average power is then estimated to be $(-2 \text{ V} \times -0.05 \mu\text{A} + 2 \text{ V} \times 50 \mu\text{A}) \div 2 \approx 50 \mu\text{W}$ for our RC system.

Figure R2 (a) Current response under different voltage pulse amplitudes (2 V ~ 3 V). It can be seen that the increase of current shows two different regimes: when the voltage is just applied, the current shows a very fast increase and then the current increase becomes much slower after a few microseconds. Thus, the experimental results can be well fitted with a double exponential function shown as: $I = I_\infty - I_a \exp\left(-\frac{t}{t_a}\right) - I_b \exp\left(-\frac{t}{t_b}\right)$, where $t_a(t_b)$, $I_a(I_b)$, and I_∞ represent the characteristic time, the prefactor, and a current constant for the fast-increase (slow-increase) process, respectively. (b) The characteristic time t_a , which determines the fastest pulse width we can apply on the device, decreases with the increasing applied voltage. When the applied voltage is greater than 2.4 V, t_a is less than 10 μs . That means the pulse width (120 μs) of the input signal in our RC system can be further reduced to 10 μs or below, which would greatly reduce the system latency.

Comment #3

(3) The spoken digit classification is a useful example to demonstrate the accuracy of the memristor RC system. However, Fig 3(d) does not convey meaningful information about the accuracy because the color scale ends at 50% correct and clearly all of the data variation is in the range of above 50%. I would recommend rescaling or removing that plot.

The example of MNIST being performed on the RC net is of much less interest, as handwritten images are not a dynamic system that requires an RC net, and this is a relatively easy and efficient system to solve on a standard multi-layer perceptron or convolutional neural network. I would recommend replacing this with a more relevant example.

Response:

Thank you for your comment. We would like to clarify that the color bar in Fig. 3d actually represents the number of predicted digits under the correct value, rather than the accuracy rate. **The “50” in the color bar represents the max number of tested digits for one correct value.** To avoid misunderstanding, **we have normalized the color bar in the revised manuscript** as shown in **Figure R3** below. Furthermore, as the reviewer suggested, we have carried out new experiments and replaced the original handwritten-digit recognition task (MNIST) with a more sophisticated task, time series prediction. **The new task is to reconstruct the Hénon map, which is a widely used benchmark task in the field of RC research** (e.g. Rodan A. et al., *IEEE Trans. Neural Netw.*, 2011; Sun X. et al., *IEEE Internet of Things Journal*, 2019). The results are shown in **Figure R4** (see detailed discussions below). After finding the suitable parameters, our RC system can achieve excellent performance on this task. The lowest prediction error achieved by memristor-based RC system (NRMSE = 0.046) is only half of the value achieved by a standard ESN system (NRMSE = 0.091) as reported in the previous work (e.g. Rodan A. et al., *IEEE Trans. Neural Netw.*, 2011). These new results further confirm the ability of our memristor-based RC system in handling temporal tasks with high accuracy.

To clarify this point, we have added **Figure R4** as **Figure 4** in the revised manuscript and the related discussions in the revised manuscript:

On pages 10 – 13, lines 249 – 311: “**In addition to the classification of temporal signals in the above two demonstrations, we also perform another benchmark task to demonstrate the prediction of temporal signals. Hénon map has been established as a typical discrete-time dynamic system with chaotic behavior. It describes a nonlinear two-dimensional (2-D) mapping that transforms a point $(x(n), y(n))$ on the plane into a new point $(x(n+1), y(n+1))$, defined as follows:**

$$x(n + 1) = y(n) - 1.4x(n)^2 \quad (2)$$

$$y(n + 1) = 0.3x(n) + w(n) \quad (3)$$

where $w(n)$ is a Gaussian noise with a mean value of 0 and a standard deviation of 0.05. The task is to predict the system position at time step $n + 1$, given the values up to time step n . The system can be described as an equation containing only x if we

combine Eq. (3) and Eq. (2), so the input of the task is $x(n)$ and the target output is $x(n+1)$. Using these equations, we generate the Hénon map dataset with a sequence length of 2000, in which the first 1000 data points is used for training and the rest is used for testing. To execute this task in our memristor-based parallel RC system, the input time series $x(n)$ is linearly mapped to the voltage range of $[V_{\min}, V_{\max}]$. The mask process is similar to the one used in the previous waveform classification. During each time interval τ , the pre-processed signal is multiplied by a special mask with a length of M to generate the input voltage sequence with a time step δ ($\delta = 120 \mu\text{s}$). An N -parallel RC system is realized by using different mask sequences. The training and testing processes are similar to the previous tasks and the only difference is that a bias is added to the output layer to neutralize the influence of input signal offset on the output. Both bias and weights can be obtained by training with linear regression. After finding the suitable parameters, our RC system can achieve excellent performance on the time-series prediction. For example, Fig. 4a shows the predicted time series versus the ideal target during testing process for the first 200 time steps, where a very low NRMSE of 0.046 is achieved by the dynamic memristor-based RC system. Here, the parameters are set to $M = 4$, $N = 25$, $V_{\max} = 2.5 \text{ V}$ and $V_{\min} = -0.8 \text{ V}$. In order to show the predicted results more intuitively, Fig. 4b is a 2-D display of the Hénon map in Fig. 4a. As can be seen in the figure, the strange attractor of the Hénon map can be well reconstructed.

As mentioned above, the parameter setting has a big impact on the performance of the memristor-based RC system. As shown in Fig. 4c, the output of our RC system has a relatively large prediction error with an NRMSE of 0.14, when changing M and V_{\max} to 25 and 2.0 V respectively while keeping V_{\min} and the total reservoir size ($M \times N$) the same. Furthermore, a systematic experiment is conducted and the results are shown in Fig. 4d, where the system performance varies with the two parameters of M and V_{\max} . Here the parameter V_{\max} is related to the input scaling, which has been proven to be an important parameter that affects the performance of RC system. Different input scalings are realized by simply changing V_{\max} when setting V_{\min} to be a fixed value close to 0. In the experiment, V_{\min} is empirically set to be a small negative value (-0.8 V) in order to balance the resistive state of the dynamic memristor. It can be seen from Fig. 4d that the prediction error (NRMSE) varies with not only M but also V_{\max} obviously. The best performance is achieved when $M = 4$ and $V_{\max} = 2.5 \text{ V}$, as too large or too small M and V_{\max} would cause relatively poor prediction results. Similar experimental results are obtained by testing on different devices as shown in Supplementary Information Fig. S9. The effect of mask length has been analysed in the previous sections. Here, we further study the influence of V_{\max} on the performance of the memristor-based RC system. The value of V_{\max} determines the nonlinear region of the device in response to the input signal. As shown in Supplementary Information Fig. S8a, the dynamic memristor response to the input voltage has an apparent threshold. The region around the threshold has a strong nonlinearity, while the region far away from the threshold has a weaker nonlinearity. If V_{\max} is too small, the resistance state of the device is difficult to be changed (see Supplementary Information Fig. S8b), which would lead to poor system performance. However, if

V_{\max} is too large, the overall nonlinearity in the entire input region would be reduced, which also degrades the RC system performance. Therefore, in order to achieve the best system performance, the value of V_{\max} needs to be carefully adjusted.

In addition, the comparison of the prediction error versus reservoir size between the software-based and memristor-based RC systems is shown in Fig. 4e. The lowest prediction error achieved by memristor-based RC system (NRMSE = 0.046) is only half of the value achieved by a standard ESN system (NRMSE = 0.091) as reported in previous work (Rodan A. et al., *IEEE Trans. Neural Netw.*, 2011), and the total reservoir size used in our RC system is also half of that in the standard ESN system. For comparison, the simulation result using a simple dynamic memristor model is also shown in Fig. 4e, where the prediction error achieved by simulation is slightly lower than that achieved by experiment. The simulation details are described in Fig. S10 and Table S1 in the Supplementary Information. These results suggest that the dynamic memristor-based parallel RC system that we proposed in this work still has plenty of room for performance optimization.”

Figure R3. Predicted results obtained from the memristor-based RC system versus the correct outputs, where the word error rate is as low as 0.4%. The two parameters M and N of the RC system are set to be 10 and 40 respectively. Colour bar represents the normalized probability of each predicted result under the correct output.

Figure R4. Demonstration of Hénon map prediction. (a) The predicted results obtained by the memristor-based parallel RC system, where the black line represents the ideal target and the red line represents the experimental output from the RC system. Test parameters are set to be $M = 4$, $N = 25$, $V_{\max} = 2.5$ V and $V_{\min} = -0.8$ V. (b-c) 2-D display of the predicted results under different test parameters, where the M and V_{\max} in (b) and (c) are 4, 2.5 V and 25, 2.0 V respectively. V_{\min} and the total reservoir size ($M \times N$) remains unchanged at -0.8 V and 100 respectively. (d) The prediction error varies with the two test parameters M (1 ~ 100) and V_{\max} (2.0 ~ 3.0 V), where the colour bar represents the NRMSE (the values greater than 0.13 are shown in black in the figure). V_{\min} and $M \times N$ remain unchanged. (e) The NRMSE changes with the reservoir size in different RC systems including the standard ESN (results in Ref. 40), memristor-based parallel RC and software-simulated one. In both experiment and simulation, the mask length remains constant at 4, and the total reservoir size is adjusted by changing the number of parallel reservoirs N .

Reviewer #2 (Remarks to the Author):

This paper discusses a dynamic memristor based reservoir computing (RC) in processing spatial and temporal signals by choosing a certain mask length and the number of reservoirs. The authors show the simulation results through low NRMSE and word error rate for waveform classification and spoken-digit recognition and has a high recognition rate for handwritten-digit recognition. The use of memristor in realizing this memristor-based RC is well explained.

This work also has a potential impact to the field since the RC system does not use additional read operations and uses the device nonlinearity from the memristors. This makes the system very efficient. The described system is able to handle spatial-temporal tasks very efficiently. Overall, this paper gives a solid explanation of the system with good results compared to software-based RC system and shows compact, efficient and feasible results with memristor.

Response:

We thank the reviewer for recognizing the impact of this work to the field of RC and the significance of the implementation of dynamic memristor-based RC system. Our detailed responses to your technical comments are provided below.

Some concerns are listed below:

Comment #1

(1) This Memristor-Based RC system paper states that to enable parallel processing of input signals a more sophisticated RC system can be used. Here the experiment was realized using a single memristor in multiple cycles so more information could be included for the process of making a more sophisticated RC.

Response:

Thank you very much for your comment. In our experiment, the parallel RC system is simulated by testing single memristor in multiple cycles as the reviewer mentioned. In the future, a multi-layer parallel RC system can be realized by integrating multiple memristor devices as illustrated in **Figure R5** below. Each layer is composed of a memristor-based parallel RC system. The overall reservoir states consist of the original input and the state response of each layer. The system output is a linear combination of all reservoir states. Here the variation among devices could increase the reservoir state richness and hence further improve the system performance. Compared with the single-layer RC system demonstrated in this work, **such multi-layer RC system would have an improved performance because of its richer reservoir states and larger memory capacity** (e.g. Gallicchio C. et al., *Neural Networks*, 2018).

To clarify this point, we have added **Figure R5** as **Supplementary Figure S12** in the revised **Supplementary Information**:

Figure R5. Schematic diagram of dynamic memristor-based multilayer RC system. Each layer is composed of a memristor-based parallel RC system, and the difference between layers is determined by the mask matrix. The overall reservoir states consist of the original input and the state response of each layer. The system output is a linear combination of all reservoir states. Compared with the single-layer RC system, the multi-layer RC system would have an improved performance because of its richer reservoir states and stronger memory capacity.

Comment #2

(2) The process for determining the mask length can also be discussed in detail since it is a key feature for this paper.

Response:

Thank you very much for your comment. As the reviewer pointed out, using the mask length to adjust the overall properties of the reservoir and improve the RC system performance is the key innovation of this work. The main function of the mask is to produce rich reservoir states in a single device, where the mask length determines the number of states. Taking the waveform classification task as an example, the mask process is to convert the input signal in a sampling period into a certain binary sequence. Different orders of these binary sequences would cause the difference in the amplitude of the device response, because of the dynamic and nonlinear characteristics of the device. Therefore, **instead of using only one long mask to generate the reservoir states, we can use multiple different short masks to generate the reservoir states with the same richness** (as used in the parallel RC system proposed in this paper). It also enables us to adjust the mask length freely while fixing the total reservoir size, which is critical to the improvement of system performance as discussed in the manuscript. In the adjustment of the mask length, **there is a tradeoff between the reservoir state richness and the feedback strength between two time durations as shown in Figure R6**: a longer mask length enriches the reservoir state but also weakens the feedback strength since the reservoir states would reach the upper or lower limit in the long time duration and hence lose the ability to respond to the signal in next duration. Thus, in order to achieve the best system performance, the mask length needs to be carefully adjusted to ensure both large feedback strength and sufficient state richness. The experimental results in **Figure R7** show that the best RC system performance is achieved when the mask length is 4, where the lowest classification error of NRMSE = 0.14 is obtained.

To clarify this point, we have added **Figure R6** and **Figure R7** as **Supplementary Figure S3** and **Figure S4** respectively in the revised **Supplementary Information** and also incorporated the above discussions in the revised manuscript:

On pages 7, lines 161 – 173: “**To explain such dependence on the mask length, let us consider two extreme cases with mask lengths of 40 and 1. When the mask length is as long as 40, the overall change of memristor conductance over duration τ would be large, which could easily drive the reservoir states to reach the upper or lower limit, thereby losing the ability to further process signals in the subsequent durations. In other words, the feedback strength between the two time durations decreases as the mask length increases, leading to a larger classification error. On the other hand, when the mask length is as short as 1, the binary combination of the mask sequence would be very limited, which limits the types of the mask sequence. In this case the richness of the reservoir states in the parallel RC system is very low and the effective reservoir states could not support successful classification, leading to a large classification error as well. So in order to achieve the best classification results, the mask length needs to**

be carefully adjusted to make a trade-off between the feedback strength and the state richness.”

Figure R6. (a) - (c) Waveform classification results when the mask length changes from 1 to 40. The first to fourth panels of each figure show the input waveform, the first reservoir state, all reservoir states, and the classification results, respectively. As we can see from the second panel of each figure, one of the reservoir states extracted from the dynamic memristor response can transform the difference in waveforms into the change of amplitude (ΔV_p). This effect can be described as the feedback strength between two duration τ ; however this feedback strength decreased with the increase of mask length. That is because the mask length can affect the overall change of memristor conductance over a duration τ , as we mentioned in the main text. In addition, from the third panel of each figure we can find that more and more reservoir states overlap as the mask length decreases. When the mask length is 1, only two reservoir states can be distinguished, leading to a large error rate. This result further confirms our conclusion in the main text.

Figure R7. (a) The feedback strength and state richness changes with the mask length M , where the feedback strength and state richness are quantified as $\Delta \bar{V}_p$ and $\sigma(V_{p1})$ respectively. $\Delta \bar{V}_p$ is the average of the five largest ΔV_p in all reservoir states and $\sigma(V_{p1})$ is the standard deviation of V_{p1} that is the peak response voltage of the sine wave. We can see that the feedback strength decrease with the mask length while the state richness increases with the mask length and saturates when the mask length is larger

than 4. The best performance of the RC system is achieved when the mask length is around 4 that yields a trade-off between the feedback strength and the state richness. (b) The classification error (NRMSE) changes with the mask length, and NRMSE reaches the minimum value of 0.14 when the mask length is 4. The error bars in both (a) and (b) show the variation between devices.

Comment #3

(3) An explanation could be added as to how the value of the load resistor was calculated.

Response:

Thank you for your comment. The load resistor R_L in our hardware system is used to detect the response current of dynamic memristor. The value of this resistor is dependent on the magnitude of the current response $I_{memristor}$ and the maximum gain of the amplifier ($A_v = 1000$) we used. In the speech recognition task, our system need detect a current on the order of $1 \mu\text{A}$. As the voltage upper limit of our ADC is V_{ADC} 3.3 V, the load resistor should satisfy the following equation: $\frac{V_{ADC}}{I_{memristor} \times R_L} \leq A_v$. **In our system, we have $R_L \geq \frac{V_{ADC}}{I_{memristor} \times A_v} \approx 3.3 \text{ k}\Omega$.** In addition, in order to reduce the voltage drop on the load resistor connected in series with the dynamic memristor, R_L should be much smaller than the memristor resistance ($7 \text{ M}\Omega \sim 20 \text{ k}\Omega$ measured in voltage range of 1 to 3 V). As a result, the value of R_L is chosen to be $4.7 \text{ k}\Omega$ in our experiment.

To clarify this point, we have added the above discussion in the revised manuscript: On pages 15, lines 359 – 367: “**The value of R_L is dependent on the magnitude of the current response $I_{memristor}$ and the maximum gain of the amplifier ($A_v = 1000$) we used. In the speech recognition task, our system need detect a current on the order of $1 \mu\text{A}$. As the voltage upper limit of our ADC is V_{ADC} 3.3 V, the load resistor should satisfy the following equation: $\frac{V_{ADC}}{I_{memristor} \times R_L} \leq A_v$. In our system, we have $R_L \geq \frac{V_{ADC}}{I_{memristor} \times A_v} \approx 3.3 \text{ k}\Omega$. In addition, in order to reduce the voltage drop on the load resistor connected in series with the dynamic memristor, R_L should be much smaller than the memristor resistance ($7 \text{ M}\Omega \sim 20 \text{ k}\Omega$ measured in voltage range of 1 to 3 V). As a result, the value of R_L is chosen to be $4.7 \text{ k}\Omega$ in our experiment.**”

Comment #4

(4) More information about the memristor could be added about the device characteristics of the memristor such as the critical current and the reason for the choice of elements used.

Response:

Thank you for your comment. Per your suggestion, we have added more data about the characteristics of the memristor, as shown in **Figure R8** and **Figure R9**. Here the I-V nonlinearity under different voltage sweeping ranges V_{\max} and the characteristic time of current increase under different voltage amplitudes have been further studied.

In software-based RC systems, the input scaling is an important parameter that affects the system performance, which determines the nonlinear region of system operation. Similar to that, the voltage range is used to adjust the system nonlinearity in the memristor-based RC system. As shown in **Figure R8, I-V curve of the dynamic memristor has a strong rectifying characteristic which is similar to the ReLU function widely used in artificial neural networks (ANNs)**. The ReLU function has an obvious threshold. If the input signal is close to the threshold, the output has a strong nonlinearity. On the contrary, if the input signal is much larger than the threshold, the output is almost linear. In the dynamic memristor, the rectified I-V curve has a similar property and it can be roughly divided into two parts with strong nonlinearity and weak nonlinearity respectively. **In order to guarantee enough nonlinearity for the RC system, the input voltage needs to be limited as much as possible to a small range around the threshold. Meanwhile, the input voltage should not be too small; otherwise it would be difficult to change the state of the dynamic memristor.** Considering all these factors, we find that the most suitable input voltage is in the range of -0.8 to 2.5 V (see Figure 4d in the revised manuscript), which not only guarantees the nonlinearity of RC system but also is large enough to ensure that the device state can be changed.

As shown in **Fig. 1f** of the manuscript, the characteristic time of the device conductance decay has been analyzed, and it is a very important parameter that determines the upper limit of the input pulse width. To prevent the device from saturating to a state that is independent of previous inputs, the input pulse width should be much smaller than the characteristic time (400 μs) of the conductance decay. Here, we further evaluate the lower limit of the input pulse width, which is critical for reducing the system latency. As shown in **Figure R9a**, the behavior of current increase under different voltage amplitudes has been studied. It can be seen that the increase of current shows two different regimes: when the voltage is just applied, the current shows a very fast increase and then the increase becomes much slower after a few microseconds. The characteristic time in the fast-increase regime determines the fastest pulse width we can apply on the device. In order to obtain the characteristic time, a double exponential function is used to fit the experimental data. As shown in **Figure R2b**, the characteristic time t_a decreases when the voltage amplitude

increases, and t_a becomes less than 10 μs when the applied voltage is greater than 2.4 V. That means the pulse width (120 μs) of the input signal in our RC system can be further reduced to 10 μs , which will greatly optimize the system latency.

To clarify this point, we have added **Figure R8** and **Figure R9** as **Supplementary Figure S8** and **Figure S2** respectively in the revised **Supplementary Information** and also incorporated the above discussions in the revised manuscript:

On pages 12, lines 293 – 301: “The value of V_{max} determines the nonlinear region of the device in response to the input signal. As shown in Supplementary Information Fig. S8a, the dynamic memristor response to the input voltage has an apparent threshold. The region around the threshold has a strong nonlinearity, while the region far away from the threshold has a weaker nonlinearity. If V_{max} is too small, the resistance state of the device is difficult to be changed (see Supplementary Information Fig. S8b), which would lead to poor system performance. However, if V_{max} is too large, the overall nonlinearity in the entire input region would be reduced, which also degrades the RC system performance. Therefore, in order to achieve the best system performance, the value of V_{max} needs to be carefully adjusted.”

Figure R8. (a) Typical I-V curve of the dynamic memristor during positive voltage sweep. The I-V curve can be roughly divided into two regions with strong nonlinearity and weak nonlinearity respectively. (b) The I-V loops under different voltage sweeping ranges V_{max} .

Figure R9 (a) Current response under different voltage pulse amplitudes (2 V ~ 3 V).

It can be seen that the increase of current shows two different regimes: when the voltage is just applied, the current shows a very fast increase and then the current increase becomes much slower after a few microseconds. Thus, the experimental results can be well fitted with a double exponential function shown as: $I = I_{\infty} - I_a \exp\left(-\frac{t}{t_a}\right) - I_b \exp\left(-\frac{t}{t_b}\right)$, where $t_a(t_b)$, $I_a(I_b)$, and I_{∞} represent the characteristic time, the prefactor, and a current constant for the fast-increase (slow-increase) process, respectively. (b) The characteristic time t_a , which determines the fastest pulse width we can apply on the device, decreases with the increasing applied voltage. When the applied voltage is greater than 2.4 V, t_a is less than 10 μs . That means the pulse width (120 μs) of the input signal in our RC system can be further reduced to 10 μs or below, which would greatly reduce the system latency.

Reviewer #3 (Remarks to the Author):

The authors present a dynamic memristor-based reservoir computing approach and use it for solving spatiotemporal signals. The article fails to cite key background literature, does not provide sufficient details, and most importantly, does not compare the results with related/state-of-the-art work. The only comparison the authors provide is with a software-based implementation. However, there's a significant body of work on memristor-based reservoirs today. The authors should have compared the performance of their proposed approach with the existing literature, not just with a software-based approach. Without a comparison with other memristor-based RC systems, a reader will be unable to evaluate the value and the impact of what the authors propose.

Response:

We thank the reviewer for the critical comments on our work that are very helpful to further improve the quality of this paper. In the revised manuscript, we have cited key background literature and compared our system with related work on memristor-based RC systems in the literature. Our detailed responses to your technical comments are provided below.

Some concerns are listed below:

Comment #1

(1) The concept of the "virtual nodes" is not motivated and described in detail.

Response:

Thank you for your comment. The concept of virtual nodes is proposed in the RCs based on dynamic systems, which is equivalent to the physical nodes in traditional RCs based on neural network (e.g. Appeltant L. et al., *Nat. Commun.*, 2011). In this paper, **we use the property of the reservoir state to represent the virtual nodes.** **Figure R10** shows the response current of the dynamic memristor under masked voltage pulses, and the reservoir states (or the outputs of virtual nodes) are collected by sampling the response current in a fixed time step. In order to generate rich reservoir states, the system response in a time duration τ needs to be separated sufficiently. It thus requires the system to have rich dynamics as shown in **Figure R10c-d**, where the system response can be well separated when the time step δ is smaller than the characteristic time of the dynamic memristor. **In this way, the virtual nodes are coupled with each other through the system dynamics, which corresponds to the random connections of physical nodes in traditional RCs.**

To clarify this point, we have added **Figure R10** as **Supplementary Figure S1** in the revised **Supplementary Information**:

Figure R10. Investigation of the effect of the time step on the virtual nodes of the RC system. (a) The input voltage after the mask process and the corresponding current response of the dynamic memristor, where the time step δ (50 ms) is much larger than the characteristic time t_0 (400 μ s) of the device. In this case, the device rapidly saturates to a state that is independent of previous inputs. (b) As a result, the extracted virtual nodes are independent of each other, and each node is only coupled with itself at the previous time step. (c) The input voltage after the mask process and the corresponding current response of the dynamic memristor, where the time step δ (50 μ s) is smaller than the characteristic time t_0 (400 μ s) of the device. In this case, the device does not have enough time to reach a saturated state. (d) As a result, the extracted virtual nodes can be coupled with their neighbours efficiently, and hence a functional RC system can be implemented.

Comment #2

**(2) The authors fail to cite the first memristive RC paper from 2012:
<https://ieeexplore.ieee.org/document/6464167>**

Response:

Thank you very much for letting us know this key literature work, which is indeed the first paper proposing the concept of memristor-based RC system. We have now cited it as ref. 24 in the revised manuscript:

On pages 19, lines 449 – 450: “*24. Kulkarni M. S. & Teuscher C., in 2012 IEEE/ACM International Symposium on Nanoscale Architectures (NANOARCH), 226-232 (IEEE, 2012).*”

Comment #3

(3) "64: [...] to control the variation between devices and hence it is not well reproducible." Well, isn't the goal of variation precisely to provide rich dynamics that do not need to be reproducible? That's the entire point of RC. The training of the output layer will deal with that "problem," which is not really a problem. In fact, it's generally good to have variation. See for example: <https://ieeexplore.ieee.org/document/6623028/>

Abstract:

- "[...] however, this method is not well controllable and reproducible."  It is not clear why the authors are saying that. This should be clarified even in the abstract.

Response:

Thank you for your comment. We apologize for the misleading statement in the abstract, which was also pointed out by the first reviewer. We agree with the reviewer that device variation is important in the implementation of memristor-based RC system, and we have cited the work <https://ieeexplore.ieee.org/document/6623028/> as ref. 33 in the revised manuscript. Here we intend to emphasize that **many critical parameters of the RC system, such as the state richness and feedback strength of the reservoir, cannot be further adjusted in the previous work of ref. 22.** These parameters of the reservoir have significant impacts on the RC system performance (e.g. Appeltant L. et al., *Nat. Commun.*, 2011; Larger L. et al., *Optics Express*, 2012), so if they cannot be adjusted, then there could be a situation in which no matter how we train the output weights, the system still has a poor performance. Besides, the desired values of these parameters generally depend on the specific tasks. Therefore, **it is necessary to represent these parameters using the system properties that can be externally adjusted**, similar to the case in software-based RC systems where parameters like the input weight scaling, reservoir sparsity and spectral radius of the weight matrix are usually selected and optimized to represent the overall properties of the reservoir (e.g. Rodan A. et al., *IEEE Trans. Neural Netw.*, 2011; Gallicchio C. et al., *Neurocomputing*, 2018; Jaeger H. et al., *GMD-Forschungszentrum Informationstechnik*, 2002., 2002). Inspired by this idea, in this work **we proposed to use the mask length and input signal range as adjustable parameters to further tune the properties of the reservoir and achieve the best RC system performance, as experimentally demonstrated in Figure R11 below.** The device-to-device variation represented by the error bar in **Figure R11** also proves that our RC system is reproducible when using different dynamic memristors for the system implementation. By controlling the parameters in the mask process, we can adjust not only the state richness of the reservoir but also the feedback strength both of which are critical properties that affect the RC system performance. **This is one of the key novelties of our work beyond ref. 22.**

To clarify this point, we have added **Figure R11** as **Supplementary Figure S4** in the

revised **Supplementary Information** and also incorporated the above discussions in the revised manuscript:

On pages 1, lines 20 – 21: “however, the state richness created by such method is fixed after the devices are prepared, and cannot be further adjusted in order to optimize the system performance.”

On pages 3, lines 64 – 70: “In a RC system, there are several properties of the reservoir that largely affect the system performance, of which the richness of the reservoir states is one of the most important parameters. In the previous works, different reservoir states were usually generated using the inherent device-to-device variations. Although this method can generate many reservoir states, the state richness is fixed after the devices are prepared, and cannot be further adjusted in order to optimize the system performance.”

On pages 3 – 4, lines 75 – 78: “By controlling the parameters in the mask process, we can adjust not only the state richness of the reservoir but also the feedback strength both of which are critical properties that affect the RC system performance.”

On pages 20, lines 468 – 469: “33. Bürger J. & Teuscher C., in *2013 IEEE/ACM International Symposium on Nanoscale Architectures (NANOARCH)*, 1-6 (IEEE, 2013).”

Figure R11. (a) The feedback strength and state richness changes with the mask length, where the feedback strength and state richness are quantified as $\overline{\Delta V_p}$ and $\sigma(V_{p1})$ respectively. $\overline{\Delta V_p}$ is the average of the five largest ΔV_p in all reservoir states and $\sigma(V_{p1})$ is the standard deviation of V_{p1} that is the peak response voltage of the sine wave. We can see that the feedback strength decrease with the mask length while the state richness increases with the mask length and saturates when the mask length is larger than 4. The best performance of the RC system is achieved when the mask length is around 4 that yields a trade-off between the feedback strength and state richness. (b) The classification error (NRMSE) changes with the mask length, and NRMSE reaches the minimum value of 0.14 as the mask length is 4. The error bars in both (a) and (b) show the variation between devices.

Comment #4

(4) The authors use many vague terms, such as "not well," almost equivalent," "for a while," etc. These must be replaced by quantitative statements.

"67. where an extremely low normalized root mean square error (NRMSE) of 0.14 and word error rate of 0.4% are achieved respectively." Compared to what?

Abstract:

- "The performance of dynamic memristor-based RC system is almost equivalent to the software-based one" -> The authors need to quantify that statement. How much better? E.g., how many %?

"37. As a result, input signal can stay in the network for a while so that short-term memory can be realized in this way"  This sentence is unclear. I intuitively know what the authors mean, but the statement needs to be rewritten and clarified.

Response:

Thank you for your comment. Following your suggestions, we have revised the vague expressions in the paper and further quantified the advantages of our RC system. Specifically, in the waveform classification task, the lowest NRMSE we have achieved is 0.14 which is lower than the previous value of 0.2 obtained with spintronic oscillator-based RC system (e.g. Torrejon J. et al., *Nature*, 2017). In the spoken-digit recognition task, the lowest word error rate we get is 0.4% which is lower than the value of 0.8% obtained by the memristor-based RC system in previous work (e.g. Moon J. et al., *Nat. Electron.*, 2019). Also, in the Hénon map prediction task, the lowest prediction error achieved by our RC system (NRMSE = 0.046) is only half of the value achieved by a standard ESN system (NRMSE = 0.091) as reported in previous work (e.g. Rodan A. et al., *IEEE Trans. Neural Netw.*, 2011).

We have further estimated the power consumption of our RC system. In this work, because of the mask process, both the input voltage and the corresponding current vary within a certain range. Thus, the power consumption of processing a single input can be estimated by calculating the average power for all the inputs in a task. For the spoken-digit recognition task, the input voltage mostly varies from -2 to 2 V (the corresponding current is in the range of -0.05 to 50 μ A), so **the average power is then roughly estimated to be** $(-2 \text{ V} \times -0.05 \text{ } \mu\text{A} + 2 \text{ V} \times 50 \text{ } \mu\text{A}) \div 2 \approx 50 \text{ } \mu\text{W}$ **for our RC system.** In the previous work (e.g. Moon J. et al., *Nat. Electron.*, 2019), the power consumption for processing a single input is approximately $3.0 \text{ V} \times 100 \text{ } \mu\text{A} = 300 \text{ } \mu\text{W}$, which is 6 times higher than the value estimated in this work. A brief comparison is made in **Table R2**.

To clarify this point, we have added **Table R2** as **Supplementary Table S2** in the revised **Supplementary Information** and also incorporated the above discussions in the revised manuscript:

On pages 2, lines 47 – 49: “As a result, the history information of the input signal can be encoded into the internal states of the network so that short-term memory can be realized in this way.”

On pages 7, lines 173 – 175: “In experiment, we find the optimal mask length to be around 4 that yields the lowest NRMSE of 0.14, which is lower than the previous value of 0.2 obtained with spintronic oscillator.”

On pages 10, lines 236 – 239: “The word error rate is as low as 0.4% (i.e., recognition rate of 99.6%) when M and N are set to be 10 and 40 respectively, which is lower than the value of 0.8% obtained by the memristor-based RC system in previous work.”

On pages 12, lines 303 – 307: “The lowest prediction error achieved by our dynamic memristor-based RC system (NRMSE = 0.046) is only half of the value achieved by a standard ESN system (NRMSE = 0.091) as reported in previous work, and the total reservoir size used in our RC system is also half of that in the standard ESN system.”

RC System	Power per input	Performance
Previous work (ref. 22)	300 μ W	99.2%
This work	50 μ W	99.6%

Table R2. The estimated power and performance of different memristor-based RC systems in the spoken-digit recognition task. In previous work (e.g. Moon J. et al., *Nat. Electron.*, 2019), the power consumption for processing a single input is approximately $3.0 \text{ V} \times 100 \text{ } \mu\text{A} = 300 \text{ } \mu\text{W}$, where the values of voltage and current are both taken from the reference. In this work, because of the mask process, both the input voltage and the corresponding current vary within a certain range. Thus, the power consumption of processing a single input should be obtained by calculating the average power for all the inputs in a task. For the spoken-digit recognition task, the values of input voltage mostly vary from -2 to 2V (The corresponding current is in the range of -0.05 to 50 μ A), so the average power is then estimated to be $(-2 \text{ V} \times -0.05 \text{ } \mu\text{A} + 2 \text{ V} \times 50 \text{ } \mu\text{A}) \div 2 \approx 50 \text{ } \mu\text{W}$ for our RC system.

Reference:

1. Appeltant L. *et al.* Information processing using a single dynamical node as complex system. *Nat. Commun.* **2**, 468 (2011).
2. Larger L. *et al.* Photonic information processing beyond Turing: an optoelectronic implementation of reservoir computing. *Optics Express* **20**, 3241-3249 (2012).
3. Rodan A. & Tino P. Minimum complexity echo state network. *IEEE Trans. Neural Netw.* **22**, 131-144 (2011).
4. Gallicchio C., Micheli A. & Silvestri L. Local Lyapunov exponents of deep echo state networks. *Neurocomputing* **298**, 34-45 (2018).
5. Jaeger H. Tutorial on training recurrent neural networks, covering BPPT, RTRL, EKF and the echo state network approach. *GMD-Forschungszentrum Informationstechnik, 2002.* **5**, (2002).
6. Moon J. *et al.* Temporal data classification and forecasting using a memristor-based reservoir computing system. *Nat. Electron.* **2**, 480-487 (2019).
7. Sun X. *et al.* ResInNet: A Novel Deep Neural Network With Feature Reuse for Internet of Things. *IEEE Internet of Things Journal* **6**, 679-691 (2019).
8. Gallicchio C., Micheli A. & Pedrelli L. Design of deep echo state networks. *Neural Networks* **108**, 33-47 (2018).
9. Torrejon J. *et al.* Neuromorphic computing with nanoscale spintronic oscillators. *Nature* **547**, 428-431 (2017).

REVIEWER COMMENTS

Reviewer #1 (Remarks to the Author):

This manuscript is significantly improved over the previous version. The improvements over previous state of the art have been better articulated, and the Henson mapping task is more uniquely suited to this RC network. I recommend publication after considering the following minor revisions:

1. The power analysis provided in the new Table S2 provides a valuable comparison to the work of Moon et al. However, I believe the previous work had a different latency for the same recognition task for the voltage/current analyzed. In this case, it is appropriate to include a comparison of the total energy as well. For example, if the competing task is completed several times faster, even with the higher power draw, then the total energy might be lower. Total energy for a task can be as important as the instantaneous power during the task.
2. The error of your system is found to be "half of the value achieved by a standard ESN system." Can you please specify if the baseline system and error you referenced (Rodan et al, 2011) is considered state of the art in terms of the error achieved? It is ok if it is not, because the claim of the paper is that you can achieve good results which can be further optimized with a novel hardware system. If you are claiming to have achieved better than state of the art error results on the Henon map reconstruction the community working on that class of RC algorithms will want to better understand why that is.
3. Please proofread and correct all grammar errors before publication.

Reviewer #2 (Remarks to the Author):

The revised paper was able to address the concerns and suggestions with good supplement figures and added information from the comments provided.

- The addition of the Supplementary Figure S12 and the further emphasis on how the multi-layer RC system would have improved performance because of richer states and larger capacity was helpful in clarifying the previous claim that was made on why the use of a multilayer system can provide enhanced performance.
- The revised explanation on identifying the mask length is much clearer with the addition of Figure S4.
- The added calculation of the value of load resistor was useful and the addition of the time-series prediction section and the further analysis on the influence of voltage on the performance on the memristor-based RC system was useful in understanding how the voltage range can be used to adjust the nonlinearity.

Reviewer #3 (Remarks to the Author):

Thank you for revising the manuscript. You have addressed my concerns.

I still think that the paper overall is quite incremental compared to previous work and that both the novelty and the results make a minor contribution to the field.

Response Letter to Reviewers' Comments

Reviewer #1 (Remarks to the Author):

This manuscript is significantly improved over the previous version. The improvements over previous state of the art have been better articulated, and the Henson mapping task is more uniquely suited to this RC network.

Response:

We thank the reviewer for the endorsement of our revised manuscript. Our detailed responses to your technical comments are provided below.

Some concerns are listed below:

Comment #1

(1) The power analysis provided in the new Table S2 provides a valuable comparison to the work of Moon et al. However, I believe the previous work had a different latency for the same recognition task for the voltage/current analyzed. In this case, it appropriate to include a comparison of the total energy as well. For example, if the competing task is completed several times faster, even with the higher power draw, then the total energy might be lower. Total energy for a task can be as important as the instantaneous power during the task.

Response:

Thank you for your comment. We agree with the reviewer that it is important to compare both power and energy consumption when the latencies of the two systems are different. The input voltage pulse width used in the current system is 120 μs , so the energy consumption per input can be estimated as $50 \mu\text{W} \times 120 \mu\text{s} = 6 \text{ nJ}$. In comparison, the energy consumption per input for the previous work of Moon et al. (ref. 22) is estimated to be $300 \mu\text{W} \times 10 \mu\text{s} = 3 \text{ nJ}$. As we explained in the previous response letter, the 120 μs pulse width used in our experiment is limited by our measurement hardware, and it can be further reduced to below 10 μs , which then reduces the energy consumption down to less than 0.5 nJ. Therefore, there is still plenty of room for improvement in the energy efficiency of our system.

To clarify this point, we have revised Table S2 in the Supplementary Information as Table R1:

RC System	Power per input	Energy per input	Performance
Previous work (ref. 22)	300 μW	3 nJ	99.2%
This work	50 μW	6 nJ	99.6%

Table R1. The estimated power and performance of different memristor-based

RC systems in the spoken-digit recognition task. In previous work (e.g. Moon J. et al., *Nat. Electron.*, 2019), the power consumption for processing a single input is approximately $3.0 \text{ V} \times 100 \text{ } \mu\text{A} = 300 \text{ } \mu\text{W}$, where the values of voltage and current are both taken from the reference. For the input pulse width in previous work is $10 \text{ } \mu\text{s}$, the energy consumption is then estimated as $300 \text{ } \mu\text{W} \times 10 \text{ } \mu\text{s} = 3 \text{ nJ}$. In this work, because of the mask process, both the input voltage and the corresponding current vary within a certain range. Thus, the power consumption of processing a single input should be obtained by calculating the average power for all the inputs in a task. For the spoken-digit recognition task, the values of input voltage mostly vary from -2 to 2V (The corresponding current is in the range of -0.05 to $50 \text{ } \mu\text{A}$), so the average power is then estimated to be $(-2 \text{ V} \times -0.05 \text{ } \mu\text{A} + 2 \text{ V} \times 50 \text{ } \mu\text{A}) \div 2 \approx 50 \text{ } \mu\text{W}$ for our RC system. The pulse width used in the current system is $120 \text{ } \mu\text{s}$ (limited by measurement hardware), so the energy consumption per input pulse can be estimated as $50 \text{ } \mu\text{W} \times 120 \text{ } \mu\text{s} = 6 \text{ nJ}$. Although the value is larger than that estimated in the previous work ($300 \text{ } \mu\text{W} \times 10 \text{ } \mu\text{s} = 3 \text{ nJ}$), there is still plenty of room for improvement in energy efficiency by reducing the input voltage pulse width (less than $10 \text{ } \mu\text{s}$) in future RC systems.

Comment #2

(2) The error of your system is found to be “half of the value achieved by a standard ESN system.” Can you please specify if the baseline system and error you referenced (Rodan et al, 2011) is considered state of the art in terms of the error achieved? It is ok if it is not, because the claim of the paper is that you can achieve good results which can be further optimized with a novel hardware system. If you are claiming to have archived better than state of the art error results on the Henon map reconstruction the community working on that class of RC algorithms will want to better understand why that is.

Response:

Thank you for your comment. The baseline system used in this work is a standard single-layer ESN and the result of NRMSE = 0.091 in the reference (Rodan et al, 2011) is indeed state-of-the-art performance for a single-layer RC system. Of course the prediction error can be further reduced (e.g. NRMSE = 0.0036) when using multi-layer RC system with more complex training process (e.g. Sun X. et al., *IEEE Internet of Things Journal*, 2019), which may not be fair to compare with the single-layer RC system implemented in our work. As the reviewer pointed out, our results can be further optimized with a novel hardware system, and our simulation shows that the minimum prediction error of our RC system is 0.01 (see Fig. 4e in the main text), which is close to the state-of-the-art result obtained by multilayer RC systems. In addition, we are currently working on the implementation of multi-layer RC system based on our dynamic memristors, and we expect much better prediction error can be achieved.

To clarify this point, we have incorporated the above discussions in the revised manuscript:

On pages 12 – 13, lines 308 – 313: “**It is worth mentioning that the prediction error of ESN used for comparison here is state-of-the-art value that a single-layer RC system can achieve, and lower error can be obtained when using multi-layer RC systems with more complex training process. For comparison, the simulation result using a simple dynamic memristor model is also shown in Fig. 4e, where the prediction error achieved by simulation is much lower than that achieved by experiment and is close to the values achieved by multi-layer RC systems.**”

Comment #3

(3) Please proofread and correct all grammar errors before publication.

Response:

Thank you for your comment. We have carefully proofread the manuscript and corrected all grammar errors (revisions are highlighted in red).

Reviewer #2 (Remarks to the Author):

The revised paper was able to address the concerns and suggestions with good supplement figures and added information from the comments provided.

- **The addition of the Supplementary Figure S12 and the further emphasis on how the multi-layer RC system would have improved performance because of richer states and larger capacity was helpful in clarifying the previous claim that was made on why the use of a multilayer system can provide enhanced performance.**
- **The revised explanation on identifying the mask length is much clearer with the addition of Figure S4.**
- **The added calculation of the value of load resistor was useful and the addition of the time-series prediction section and the further analysis on the influence of voltage on the performance on the memristor-based RC system was useful in understanding how the voltage range can be used to adjust the nonlinearity.**

Response:

Thank you very much for the valuable time you have spent reviewing our manuscript and providing insightful comments to help significantly improve the quality of our work. We are very glad to see that you are satisfied with our revision.

Reviewer #3 (Remarks to the Author):

Thank you for revising the manuscript. You have addressed my concerns.

I still think that the paper overall is quite incremental compared to previous work and that both the novelty and the results make a minor contribution to the field.

Response:

We thank the reviewer for taking time to review our manuscript and give valuable comments which help significantly improve the quality of our work. We are glad to hear that our revision has addressed the reviewer's comments. As the other two reviewers acknowledged, our work established a useful method using adjustable parameters to significantly boost the performance of dynamic-memristor RC system, which is an important contribution to the field of RC. Specifically, this work has made three key innovations:

- (1) We establish a controllable method (i.e., the mask process) to generate rich reservoir states, where the overall properties of reservoir can be adjusted by using different system parameters such as the mask length and input voltage range. The properties such as state richness and feedback strength have been proven to have a significant impact on the RC system performance.
- (2) We directly use the memristor response to the input signal as the reservoir state. Comparing to the previous RC system with additional read operation, our RC system could not only have a smaller latency but also more effectively utilize the device nonlinearity (which can be adjusted by simply changing the input range).
- (3) In addition to the standard waveform and spoken-digit classification, for the first time, we further demonstrated a time-series prediction task (see Figure 4) of reconstructing the Hénon map, which is a benchmark widely used in the field of RC research. An extremely low prediction error (NRMSE) of 0.046 is obtained in our dynamic-memristor-based RC system, which is only half of the value obtained with a standard echo state network (ESN).

Reference:

1. Sun X. *et al.* ResInNet: A Novel Deep Neural Network With Feature Reuse for Internet of Things. *IEEE Internet of Things Journal* **6**, 679-691 (2019).